# *Solve-Detect-Verify* : INFERENCE-TIME SCALING WITH FLEXIBLE GENERATIVE VERIFIER

## ABSTRACT

Complex reasoning with Large Language Models (LLMs) demands a careful balance between accuracy and computational cost. Verification, crucial for reliability, exacerbates this challenge. Existing methods often force a stark trade-off: robust process-based verifiers incur prohibitive costs due to iterative recomputation, while fast, efficient verifiers suffer from low precision. We introduce *FlexiVe* , a unified generative verifier designed to navigate this trade-off. FlexiVe dynamically allocates compute between rapid "fast thinking" and deliberative "slow thinking." A key innovation is our training strategy: we use Reinforcement Learning (GRPO) to specifically enhance the reliability of the fast mode. Remarkably, this targeted training generalizes, elevating the slow mode to state-of-the-art open-source performance. To optimally deploy *FlexiVe* , we propose the *Solve-Detect-Verify* (SDV) pipeline. SDV moves beyond static Best-of-N ranking, employing an efficient iterative refinement process that detects solution completion to curtail "overthinking" and uses *FlexiVe* 's feedback for targeted correction. Our results demonstrate significant improvements in both accuracy and efficiency. *FlexiVe* establishes a new open-source[1] state-of-the-art on ProcessBench, outperforming the much larger GenPRM-32B while requiring ∼2.3x fewer TFLOPS with 15x less training data. On the challenging AIME 2024 benchmark, the full SDV pipeline achieves 83.3% accuracy, surpassing strong baselines.

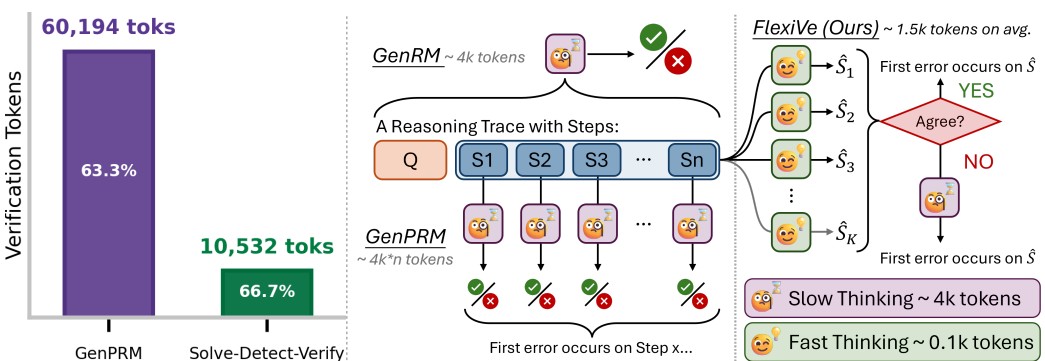

Figure 1: We introduce the *Solve-Detect-Verify* (**SDV**) pipeline, powered by our novel adaptive verifier, *FlexiVe* , to optimize the accuracy-efficiency trade-off in LLM reasoning. **Left:** On AIME 2024, our pipeline achieves higher accuracy (66.7% vs. 63.3%) while using nearly **6x fewer** verification tokens than a standard process-based approach (GenPRM). **Right:** This efficiency is driven by FlexiVe's design. Unlike Process-Based verifiers (e.g., GenPRM) that incur accumulating overhead at each step, FlexiVe analyzes the trace **holistically**. It employs a dynamic strategy: multiple low-cost "Fast Thinking" checks (∼0.1k tokens) are run first, escalating to high-cost "Slow Thinking" (∼4k tokens) only when consensus is lacking.

---

[1]Our code is available at `https://anonymous.4open.science/r/flexive-7D5D`.

# 1 INTRODUCTION

Recent advances in Large Language Models (LLMs) have enhanced their capabilities in complex reasoning tasks, primarily through the generation of step-by-step reasoning traces (Wei et al., 2022; Kojima et al., 2022). This shift towards deeper, "System 2" processes (Kahneman, 2011; Li et al., 2025; Shao et al., 2024a), while crucial for accuracy, introduces a fundamental trade-off with computational efficiency.

This challenge is exacerbated by two factors. First, models often exhibit "overthinking" (Chen et al., 2024), generating redundant self-correction steps, begins with hesitation words or phrases (e.g., "hmm", "let me double check") and redundant internal verification steps even after a correct interme- diate solution might have been implicitly reached (Chen et al., 2024). Second, ensuring the reliability of these traces requires verification (Chen et al., 2025), which adds further complexity. Sophisticated Generative Reward Models.(GenRMs) (Liu et al., 2025; Zhang et al., 2025) can be computationally prohibitive (Singhi et al., 2025), while highly efficient mechanisms like "NoThinking" (Ma et al., 2025) in Figure 3, when adapted for verification, suffer severe drops in precision (see Figure 2).

This complex interplay reveals a clear methodological gap: the need for a flexible verifier that can adapt its computational effort, and an intelligent inference-time pipeline to deploy it strategically while streamlining the reasoning process. To address these compounded challenges, we introduce FlexiVe, a unified generative verifier, and the Solve-Detect-Verify (SDV) pipeline. Our contributions are summarized as follows:

- *FlexiVe* **: A Flexible, RL-Trained Generative Verifier** We introduce a single, unified model operating across the cost-performance spectrum: (1) a rapid "fast thinking" mode; (2) a deliberative "slow thinking" mode; and (3) a dynamic "flexible" mode, which utilizes a consensus strategy that first uses efficient, parallelizable "fast thinking" assessments of the entire reasoning trace to gauge verification difficulty. It escalates to deeper, "slow thinking" analysis only when initial consensus is low. A key innovation is our training strategy: we use Group Relative Policy Optimization (GRPO) (Shao et al., 2024a;b) to specifically enhance the reliability of the "fast thinking" mode. We find this targeted RL training not only fixes the low precision of fast verifiers but generalizes remarkably, elevating the "slow thinking" mode to state-of-the-art performance.

- *Solve-Detect-Verify* **(SDV)** We propose an inference-time pipeline that intelligently inte- grates the solver and verifier, moving beyond standard Best-of-N (BoN) ranking paradigms. SDV employs an iterative refinement process, featuring a lightweight "Detect" module that leverages **likelihood-based probing** (Kadavath et al., 2022; Lin et al., 2022; Yang et al., 2024) to identify solution completion points and curtail "overthinking." Since *FlexiVe* 's feedback can be naturally used as an effective means for context engineering, the pipeline triggers *FlexiVe* to provide targeted, generative feedback that guides the solver to refine the response into a more accurate final solution. The entire detect-verify-refine cycle can be scaled; iterating the process yields accuracy gains. We demonstrate that this intelligent integration is significantly more effective than static ranking.

- **State-of-the-Art Efficiency and Accuracy** *FlexiVe* sets a new open-source SOTA on ProcessBench, outperforming larger models like GenPRM-32B while requiring ∼2.3x fewer TFLOPS and, crucially, using **15x less training data**. On challenging benchmarks like AIME 2024, the full SDV pipeline achieves 83.3% accuracy. Notably, our pipeline achieves higher accuracy than a comparable GenPRM BoN setup while using only **1/6th** of the computational tokens.

Our work demonstrates that the path to efficient and reliable LLM reasoning lies not only in developing flexible components but, critically, in designing intelligent pipelines that integrate them effectively.

# 2 RELATED WORK

**Inference-Time Scaling Strategies** Inference-time scaling strategies increase test-time compute to improve reasoning accuracy (Welleck et al., 2024; Wang et al., 2025), using methods from self- consistency (Wang et al., 2023), verifier ranked Best-of-N (BoN) (Ichihara et al., 2025), to tree-based searches (Yao et al., 2023). While effective, these strategies are computationally intensive, spurring work on optimized decoding (Sun et al., 2024) and compute trade-offs (Wu et al., 2025a). As scaling

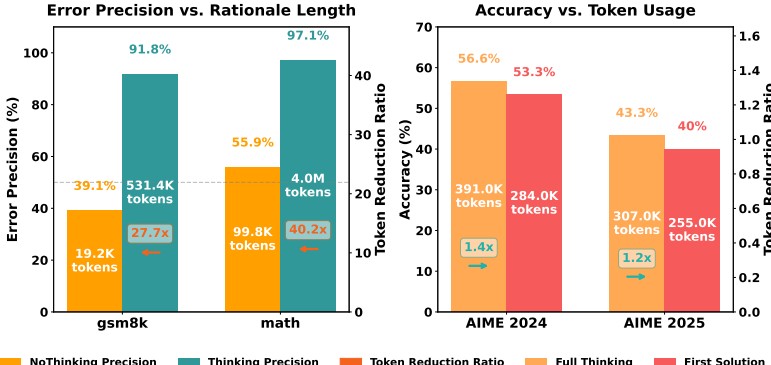

Figure 2: Empirical motivation for efficient verification and generation strategies. **(Left)** Comparison of error precision and token usage between *NoThinking* and *Thinking* verification on GSM8K and Math (ProcessBench). While *NoThinking* significantly reduces tokens, its error precision is substantially lower, sugguesting high false positive rate. **(Right)** Accuracy and token usage comparison between generating a full solution (*Full Thinking*) and halting generation early upon detecting a complete intermediate solution (*First Solution*) on AIME 2024 and AIME 2025. Early detection offers significant token reduction with comparable accuracy.

generations alone is insufficient (Chen et al., 2025) and verifier-guided search has known flaws (Wu et al., 2025b; Zhao et al., 2025a), intelligent frameworks like Solve-Detect-Verify (SDV) are needed. While building on established iterative refinement concepts (Madaan et al., 2023; Xie et al., 2023; Akyurek et al., 2023), SDV uniquely prioritizes efficiency through active detection and adaptive verification to avoid the computational redundancy ("overthinking") typical of brute-force methods (Chen et al., 2024).

**Verification Paradigms** Verification, while crucial, adds computational cost. Generative and process-based verifiers like GenRMs and PRMs (Lightman et al., 2023; Liu et al., 2025; Zhang et al., 2025) offer detailed feedback but can be demanding (Singhi et al., 2025). Recent work reduces annotation reliance via bootstrapping (Zelikman et al., 2022) or label-free methods like Math-Shepherd (Wang et al., 2024a). Hybrid models like GenPRM (Zhao et al., 2025b) integrate code execution within a process-based framework, motivating its use as a key baseline. Alternative paradigms like code-based self-verification (Zhou et al., 2024a; Wang et al., 2024b) and autoformalization (Zhou et al., 2024b) use code for precision but may lack general applicability. In contrast, FlexiVe performs efficient, holistic trace analysis with dynamic budget allocation, targeting broader use cases.

**Adaptive Computation and Our Novelty** Inspired by dual-process theory (Kahneman, 2011; Li et al., 2025), adaptive computation balances reasoning and efficiency (Graves, 2016). However, extreme efficiency methods like "NoThinking" (Ma et al., 2025) in Figure 3, when applied to verification, can yield low precision (Figure 2). The most related work, DyVe (Zhong et al., 2025), also uses "fast" and "slow" verification modes. However, its per-step approach incurs accumulating overhead. FlexiVe differs critically by (1) performing holistic, **consensus-based verification** on the entire reasoning trace to avoid iterative costs, and (2) optimizing its "fast" mode for reliable diagnosis via Reinforcement Learning (GRPO) (Shao et al., 2024a) for a more robust efficiency-accuracy balance.

---

**NoThinking**

User Query / Problem
↓
`<beginning_of_thinking>`
↓
*"Okay, I think I have finished thinking."*
↓
`</end_of_thinking>`
↓
Final Answer Generation

Figure 3: The *NoThinking* mechanism bypasses explicit thought generation, using a template to fill the thinking phase.

## 3 METHOD

### 3.1 PROBLEM FORMULATION

**System Components** Our inference-time scaling framework uses two primary Large Language Model (LLM) components: a solver LLM and *FlexiVe*, our specialized generative verifier. Both are reasoning-capable models. The solver, an off-the-shelf LLM, generates initial candidate solutions. *FlexiVe* is specifically trained for verification, detailed in Section 3.2.

**Reasoning Trace Segmentation** A reasoning trace $S_{trace}$ is parsed into an ordered sequence of $N_s$ steps, $S_{trace} = (step_1, \ldots, step_{N_s})$. Each $step_i$ is a contiguous text segment delineated by predefined "hesitation keywords" (e.g., "Wait, double-check", "Alternatively", "Hmm", "Let me check" listed in Appendix A.1.3 Figure 8). This segmented trace forms the input for verification.

**Verifier Architectures and Operation** The task of the verifier is to assess the correctness of $S_{trace}$. Architectures approach this differently, with significant implications for efficiency (Figure 1).

**Process-Based Verifiers** (e.g., GenPRM) conduct sequential, step-by-step verification. At each step $t$, the model must re-process the context of previous steps $(1 \ldots t-1)$. This growing context leads to significant computational overhead, especially for long traces.

**Holistic Verifiers** (e.g., standard GenRM and *FlexiVe* ) evaluate the entire trace in a single pass. This is inherently more efficient as the context size is fixed. While standard GenRMs often use a fixed, high computational budget, *FlexiVe* employs a dynamic strategy (Section 3.2) to modulate its effort.

In our framework, $S_{trace}$ is formatted using a critic template Zheng et al. (2024a) as input for *FlexiVe* . It outputs $V_{out} = (F, idx_{pred})$, where $F$ is a textual error analysis and $idx_{pred}$ is the predicted index of the first error ($idx_{pred} = -1$ signifies no errors). We employ a generative approach where the model articulates reasoning ($F$) rather than just outputting a scalar probability. This is crucial for two reasons: (1) Generative verification generally yields better performance by forcing the model to articulate dependencies, providing richer supervision than scalar discriminators (Liu et al., 2025; Zhang et al., 2025); and (2) in the context of our *Solve-Detect-Verify* pipeline, the textual feedback ($F$) serves as actionable diagnostic information to guide the solver during the refinement stage, which a simple scalar score cannot provide.

### 3.2 *FlexiVe* : A Unified Generative Verifier

*FlexiVe* is a unified generative verifier designed to operate across the entire spectrum of cost-performance trade-offs by leveraging a single model with three distinct inference-time modes. At one extreme, its **Fast Thinking (NoThinking)** mode, inspired by the "NoThinking" mechanism (Ma et al., 2025), this mode prioritizes extreme efficiency. It utilizes a specific template (see Figure 3) to bypass explicit thought generation, filling the thinking phase with a placeholder before directly outputting the verification result. This approach results in responses that are approximately $40\times$ shorter than the "Slow Thinking" mode (see Figure 2), enabling high-throughput, parallel sampling with minimal latency. At the other, the **Slow Thinking (Think)** mode generates a full, detailed reasoning trace to maximize verification accuracy. Our novel **Flexible Allocation (Flex)** mode dynamically bridges these approaches, adaptively switching between Fast and Slow Thinking based on perceived task difficulty to optimally balance accuracy and cost.

**Reinforcement Training for Reliable Fast Thinking** A critical challenge for efficient verifiers is their low precision (Figure 2) under NoThinking mode (Figure 3). We address this through a targeted Reinforcement Learning strategy using Group Relative Policy Optimization (GRPO) (Shao et al., 2024a;b). Our goal is to maximize the reliability of the "fast thinking" mode while maintaining its efficiency.

To achieve this, we train *FlexiVe* specifically in the "fast thinking" configuration (activating the NoThinking template during training). The model predicts the index of the first error ($idx_{gt}$) or $-1$ if correct. GRPO optimizes the policy by maximizing a composite reward $R_i = R_{\text{correct}} + R_{\text{length}}$.

The correctness reward $R_{\text{correct}}$ is defined by:

$$R_{\text{correct}}(idx_{pred}, idx_{gt}) = \begin{cases} 1.0 & \text{if } idx_{pred} = idx_{gt} \\ 0.0 & \text{otherwise} \end{cases} . \tag{1}$$

To prevent the model from "reward hacking" with verbose outputs and ensure efficiency within the "fast thinking" constraint, we apply a length-based regularization term, $R_{\text{length}}$, proportional to the length $L$ of the generated response:

$$R_{\text{length}}(L) = -\lambda \cdot L. \tag{2}$$

The hyperparameter $\lambda$ (empirically set to $0.1$) is crucial to ensure the "fast thinking" mode remains token-efficient, preventing the RL policy from converging to verbose outputs that violate the efficiency goal. Training involves sampling $G$ outputs per prompt and calculating advantages relative to the group's average (Shao et al., 2024a). A key finding, explored in Section 4.3, is that this targeted RL

training not only substantially improves "fast thinking" precision but also generalizes remarkably, enhancing the accuracy of the "slow thinking" mode.

**Flexible Allocation of Verification Budget (Flex@k)** The dynamic "Flexible" mode utilizes a two-stage verification process to tailor computational effort to the difficulty of the trace.

At inference time, the process begins with an efficient, parallelizable probing stage. *FlexiVe* performs $k$ independent "Fast Thinking" runs ,utilizing the token-efficient 'NoThinking' template, on the entire reasoning trace. The decision to escalate is determined dynamically by the consensus among these runs. Each run produces an outcome consisting of the predicted error index (or -1 if correct). We measure consensus by the agreement ratio:

$$R_{\text{agreement}} = \frac{\max_i a_i}{k}, \tag{3}$$

where $a_i$ is the count of the most frequent outcome.

If the consensus is high ($R_{\text{agreement}} \geq \tau$), it signals a straightforward case, and the "Fast Thinking" result, $V_{\text{fast}}$, is accepted efficiently. If consensus is low, it indicates ambiguity, and the framework escalates to the second stage: performing $\max(1, \lceil k/8 \rceil)$ resource-intensive "Slow Thinking" runs to produce a robust final outcome, $V_{\text{slow}}$. Methodologically, 'Slow Thinking' re-processes the problem and solver responses without appending the template shown in Figure 3.

The overall verification result $V$ is:

$$V = \begin{cases} V_{\text{fast}}, & \text{if } R_{\text{agreement}} \geq \tau, \\ V_{\text{slow}}, & \text{otherwise.} \end{cases} \tag{4}$$

The consensus threshold $\tau$ and sample count $k$ are critical hyperparameters. We selected these based on a detailed sensitivity analysis (Appendix A.3.1) and Pareto frontier analysis (Section 4.3). We identify $\tau = 0.8$ and $k = 8$ (for Flex@8) as the optimal trade-off point (the "knee" of the performance curve), maximizing accuracy gains while minimizing computational overhead.

### 3.3 *Solve-Detect-Verify*

*Solve-Detect-Verify* is a framework designed to enhance LLM reasoning accuracy and efficiency through iterative refinement, moving beyond static Best-of-N ranking. It integrates three modules: Solve, Detect, and Verify/Refine. The full pipeline is summarized in Algorithm 2 in the Appendix.

**Solve** The 'Solve' stage initiates the process, wherein the solver LLM is tasked with generating an initial, step-by-step candidate solution ($S_1$) to a given problem. This stage forms the foundational attempt at problem-solving, producing a complete reasoning trace and a final answer for subsequent evaluation.

**Detect** The 'Detect' module, as illustrated in 1 continuously monitors the output for hesitation keywords (Appendix Figure 8). Upon detection, generation pauses, and the LLM assesses solution completeness via a log-probability check ($\log p(\text{Yes})$ vs. $\log p(\text{No})$). This check efficiently reuses over 90% of the generation prefix (KV cache), minimizing overhead. If deemed complete, the pipeline advances; otherwise, generation resumes. This curtails "overthinking" and enables early verification.

---

**Algorithm 1** Solve-Detect Stage of *Solve-Detect-Verify*

**Input:** Problem $P$, Solver $M_{\text{solve}}$
**Output:** Candidate Solution $S_1$
1: **procedure** SOLVEDETECT($P, M_{\text{solve}}$)
2:      $S_1 \leftarrow \emptyset$
3:      $stop\_flag \leftarrow$ false
4:      **for** $k = 1$ **to** $L_{max}$ **do**        ▷ $L_{max}$ is max length
5:          $t_k \sim M_{\text{solve}}(\cdot | P, S_1^{(k-1)})$
6:          $S_1^{(k)} \leftarrow S_1^{(k-1)} \oplus t_k$
7:          **if** $t_k = $ EOS **then**
8:             $stop\_flag \leftarrow$ true
9:          **if** $S_1^{(k)}$ ends with $kw \in \mathcal{K}_{\text{hesitation}}$ **then**
10:             $logp_{\text{Yes}} \leftarrow \log p_{M_{\text{solve}}}(\text{Yes}|\text{Prompt}_{\text{complete}}(S_1^{(k)}))$
11:             $logp_{\text{No}} \leftarrow \log p_{M_{\text{solve}}}(\text{No}|\text{Prompt}_{\text{complete}}(S_1^{(k)}))$
12:             **if** $logp_{\text{Yes}} > logp_{\text{No}}$ **then**    ▷ Compare log-probs
13:                 $stop\_flag \leftarrow$ true      ▷ Solution complete
14:          **if** $stop\_flag$ **then**
15:             **break**
16:          $S_1 \leftarrow S_1^{(k)}$
17:      **return** $S_1$

---

**Verify and Refine** The candidate solution $S_1$ is assessed by *FlexiVe*. If correct, it is accepted. Otherwise, diagnostic feedback ($F_1$) guides the solver to generate a new solution, $S_2$. This feedback loop acts as an efficient context engineering strategy to refine the model's reasoning path.

**Iterative Refinement and Scalability** The 'Verify and Refine' stages can be iterated to progressively improve the solution. The number of iterations, $T$, is a tunable parameter that creates a trade-off

between computational cost and final accuracy. As shown in Figure 5 (top-right), each iteration yields monotonic accuracy gains, allowing the framework's computational depth to be scaled according to specific performance and budget requirements.

# 4 EXPERIMENTS

Our experiments address four primary questions: (1) How accurate and sample-efficient is *FlexiVe* compared to state-of-the-art (SOTA) verifiers? (2) Does *FlexiVe* offer a superior accuracy-efficiency trade-off (Pareto frontier) when measured in TFLOPS? (3) Does the *Solve-Detect-Verify* outperform standard inference-time scaling strategies like Best-of-N (BoN) ranking? (4) Are all components of the pipeline necessary and robust?

## 4.1 EXPERIMENTAL SETUP

For detailed experimental configurations, including hyperparameter settings and full dataset statistics, please refer to Appendix A.1.1.

**Evaluation Tasks and Datasets** We assess *FlexiVe* 's step-level verification capability (F1 score) on the comprehensive ProcessBench benchmark (Zheng et al., 2024a) (GSM8K, MATH, Olympiad-Bench, OmniMATH). For the full *Solve-Detect-Verify* , we evaluate end-to-end task accuracy and efficiency on challenging mathematical datasets: AIME (2024, 2025) (Aim, 2024; 2025), AMC, CNMO (Liu et al., 2024), and OlympiadBench. Efficiency is measured using total generated tokens and, crucially, estimated TFLOPS[2] to account for architectural differences across baselines.

**Baselines** On ProcessBench, *FlexiVe* is compared against established Process Reward Models (PRMs) (Zheng et al., 2024a), including GenPRM (7B and 32B) (Zhao et al., 2025b). For evaluating the *Solve-Detect-Verify* , DeepSeek-R1 14B (DS14B) and 32B models (Shao et al., 2024a) serve as the base "worker" LLMs. Performance is benchmarked against direct output, Self-Consistency (Majority Voting) (Wang et al., 2023), and BoN ranking using external verifiers.

*FlexiVe* **Training** *FlexiVe* (14B) is initialized from DeepSeek-R1-Distill-Qwen-14B and trained using GRPO on the BIG-Bench Mistake dataset (Tyen et al., 2024). Notably, training utilized only 1,526 samples. The training focused specifically on the 'fast mode' (NoThinking mechanism activated) to optimize rapid, reliable error detection.

Table 1: ProcessBench results reported with F1 scores. Results for *FlexiVe* are highlighted . **bold** indicates the best in the sub category. All *FlexiVe* variants are trained on only 1526 samples.

| Model | # Samples | GSM8K | MATH | Olympiad Bench | Omni-MATH | Avg. |
|---|---|---|---|---|---|---|
| *Proprietary Models* | | | | | | |
| GPT-4o-0806 | unk | 79.2 | 63.6 | 51.4 | 53.5 | 61.9 |
| o1-mini | unk | 93.2 | 88.9 | 87.2 | 82.4 | 87.9 |
| *Open Source Models (7-8B)* | | | | | | |
| Qwen2.5-Math-PRM-7B | ∼344K | 82.4 | 77.6 | 67.5 | 66.3 | 73.5 |
| RetrievalPRM-7B | 404K | 74.6 | 71.1 | 60.2 | 57.3 | 65.8 |
| Universal-PRM-7B | unk | 85.8 | 77.7 | 67.6 | 66.4 | 74.3 |
| Direct Generative PRM-7B | 23K | 63.9 | 65.8 | 54.5 | 55.9 | 60.0 |
| GenPRM-7B w/ Code Exec (Pass@1) | 23K | 78.7 | 80.3 | 72.2 | 69.8 | 75.2 |
| GenPRM-7B w/ Code Exec (Maj@8) | 23K | 81.0 | 85.7 | 78.4 | 76.8 | 80.5 |
| *Open Source Models (14-32B) w/ Moderate Compute* | | | | | | |
| Dyve-14B | 117K | 68.5 | 58.3 | 49.0 | 47.2 | 55.8 |
| GenPRM-32B w/o Code Exec (Maj@8) | 23K | 78.8 | **85.1** | 78.7 | 74.9 | 79.3 |
| *FlexiVe* (Flex@32) | **1526** | 82.8 | 83.3 | 79.2 | 73.4 | 79.7 |
| *FlexiVe* (Flex@128) | **1526** | 83.0 | 85.0 | **80.0** | **75.2** | **80.8** |
| *Open Source Models (14-32B) w/ High Compute* | | | | | | |
| GenPRM-32B (Pass@1) w/ Code Exec | 23K | 83.1 | 81.7 | 72.8 | 72.8 | 77.6 |
| GenPRM-32B (Maj@8) w/ Code Exec | 23K | 85.1 | 86.3 | 78.9 | 80.1 | 82.6 |
| *FlexiVe* (Think@64) | **1526** | **88.1** | **90.1** | **86.7** | **80.4** | **86.3** |

[2]Calculated as (input + output tokens) × model parameters, normalized.

*FlexiVe* **Configurations** We evaluate *FlexiVe* in three distinct configurations, where $k$ denotes the number of verification samples: (1) **Think@k:** Fixed "slow" budget. Performs $k$ independent "slow thinking" (deliberative) runs with a majority vote. (2) **NoThinking@k:** Fixed "fast" budget. Performs $k$ independent, token-efficient "fast thinking" runs with a majority vote. (3) **Flex@k:** Adaptive budget. Begins with $k$ "fast thinking" runs and escalates to "slow thinking" only if initial consensus is below threshold $\tau$. Provides a dynamic trade-off.

## 4.2 FLEXIVE: A UNIFIED, STATE-OF-THE-ART VERIFIER

We first evaluate the verification capabilities of the *FlexiVe* model on the ProcessBench benchmark and analyze the effectiveness of its novel RL training strategy.

**State-of-the-Art Open-Source Verification Accuracy** Table 1 details the F1 scores across various mathematical reasoning datasets. In the "High Compute" setting, *FlexiVe* (Think@64) establishes a new state-of-the-art for open-source models, achieving an average F1 score of 86.3%. This notably outperforms the compute-intensive GenPRM-32B (Maj@8) augmented with code execution (82.6% Avg F1). In the "Moderate Compute" setting, the adaptive *FlexiVe* (Flex@128) achieves a strong average F1 score of 80.8%, surpassing GenPRM-32B (Maj@8) without code execution (79.3% Avg F1).

### 4.2.1 SAMPLE EFFICIENCY AND TRAINING STRATEGY ABLATION

A key advantage of *FlexiVe* is its sample efficiency and the robustness of its RL-based training objective. To isolate the contributions of our method from the underlying base model and data size, we conducted a rigorous ablation study wirh ProcessBench (Table 2).

**RL vs. SFT on Identical Data:** We trained baselines using the exact same dataset (BIG-Bench-Mistake, 1,526 samples) and base model (DeepSeek-R1-14B). As shown in the middle section of Table 2, standard Discriminative PRM training failed to generalize (12.9% Avg F1). Supervised Fine-Tuning (SFT) on the same data reached only 49.0% Avg F1. Notably, our RL strategy not only outperformed these baselines but also surpassed an SFT model trained on $6.5\times$ more synthetic data. This confirms that the performance gains stem from the novel GRPO training strategy rather than data scale.

**Base Model Selection** We further validated our choice of base model. As shown in the top section of Table 2, while DeepSeek-R1-14B is a strong starting point (70.8% Avg F1), other open weights models like Llama-3-8B-Instruct and QwQ-32B-Preview lack the inherent reasoning capabilities required for effective verification. Crucially, *FlexiVe* significantly elevates the performance of the DeepSeek base model (from 70.8% to 75.6%), demonstrating that the gains are not merely inherited from the foundation model but are a result of our targeted alignment.

Table 2: **Ablation Study on Base Models and Training Strategies. Top:** Comparison of base models (Think@1). **Middle:** Comparison of training methods on identical data (1.5K samples). **Bottom:** Comparison of RL impact across different inference modes. The base model fails to adapt to the efficient "NoThink" and "Flex" protocols, whereas our RL training yields massive gains (e.g., +21.5% in NoThink mode).

| Model / Configuration | Training Method | # Samples | GSM8K | MATH | Olym. | Omni. | Avg. |
|---|---|---|---|---|---|---|---|
| *Base Model Selection (Think@1)* | | | | | | | |
| Meta-Llama-3-8B-Instruct | None (Base) | - | 26.8 | 13.2 | 12.3 | 13.2 | 16.4 |
| QwQ-32B-Preview | None (Base) | - | 75.5 | 59.2 | 35.7 | 35.3 | 51.4 |
| DeepSeek-R1-14B | None (Base) | - | 77.6 | 76.2 | 65.6 | 64.0 | 70.8 |
| **FlexiVe (Think@1)** | **RL (Ours)** | **1.5K** | **82.6** | **80.3** | **73.1** | **66.3** | **75.6** |
| *Training Strategy Ablation (Base: DeepSeek-R1-14B)* | | | | | | | |
| Discriminative PRM | Math-Shepherd | 1.5K | 15.8 | 15.9 | 8.3 | 11.9 | 12.9 |
| Discriminative PRM | SFT | 1.5K | 66.3 | 56.0 | 36.1 | 37.7 | 49.0 |
| Generative Verifier | SFT | 10K | 71.9 | 69.0 | 59.7 | 47.9 | 62.1 |
| **FlexiVe (NoThink)** | **RL (Ours)** | **1.5K** | **82.6** | **80.3** | **73.1** | **66.3** | **75.6** |
| *RL Impact Across Inference Modes (Base: DeepSeek-R1-14B)* | | | | | | | |
| Base Model (Flex@4) | None | - | 57.9 | 62.8 | 59.6 | 59.5 | 60.0 |
| **FlexiVe (Flex@4)** | **RL (Ours)** | **1.5K** | **78.4** | **77.7** | **72.4** | **67.3** | **74.0** |
| Base Model (NoThink@4) | None | - | 39.5 | 36.0 | 33.9 | 39.0 | 37.1 |
| **FlexiVe (NoThink@4)** | **RL (Ours)** | **1.5K** | **66.8** | **61.3** | **53.8** | **52.5** | **58.6** |

**Generalization of RL Training Across Inference Modes** A crucial finding is that our RL training instills robust verification capabilities across *all* computational budgets, not just the standard "Think" mode. We extended our ablation study (Table 2, bottom) to evaluate the base model (DeepSeek-R1-14B) acting as a verifier under our "Flex" and "NoThink" protocols. It struggles significantly with the token-efficient "NoThink" template, achieving only 37.1% Avg F1. This confirms that standard reasoning models do not inherently possess the ability to verify efficiently without dedicated alignment. In contrast, *FlexiVe* (NoThink) achieves 58.6% Avg F1, a relative improvement of ∼58%. This "fast-thinking" reliability is what powers the adaptive "Flex" mode, where *FlexiVe* outperforms the base model by 14 percentage points (74.0% vs 60.0%). Thus, our RL strategy does not merely improve reasoning for verification. It unlocks a new, efficient inference mode that the base model lacks.

### 4.3 PARETO FRONTIER ANALYSIS: ACCURACY AND EFFICIENCY

We analyze the accuracy-efficiency trade-off of *FlexiVe* against GenPRM, evaluating its Pareto frontier dominance in both theoretical TFLOPS and empirical wall-clock time. The performance gap stems from key architectural differences: *FlexiVe* is a **holistic verifier** that processes traces in a single pass, while GenPRM is a **process-based verifier** that re-evaluates an expanding context at each step, leading to non-linear cost scaling.

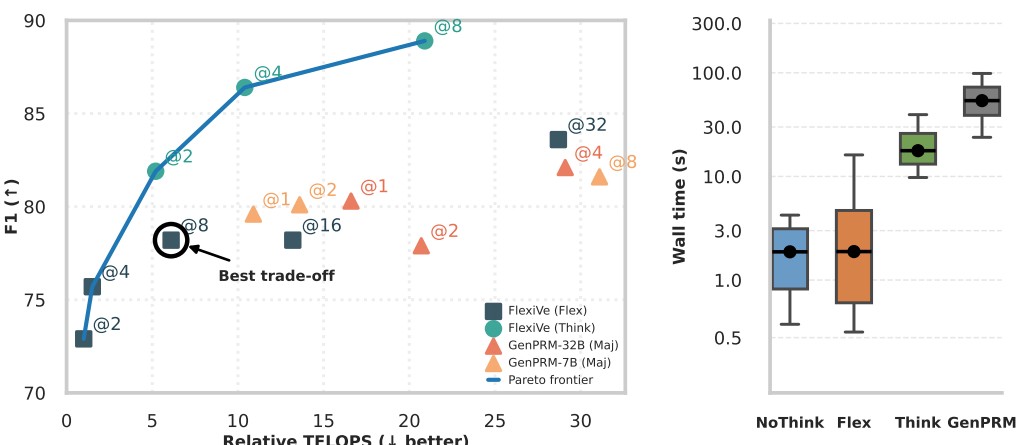

Figure 4: Pareto frontier analysis on ProcessBench MATH split. **(Left)** F1 Score versus Relative TFLOPS. *FlexiVe* (Think@k) establishes the state-of-the-art frontier, achieving higher F1 scores at a lower computational cost relative to GenPRM-7B and GenPRM-32B. **(Right)** A comparison of wall-clock time. *FlexiVe* demonstrates substantially lower latency; its Flex mode is comparable to the NoThink baseline, while its Think mode is approximately 2.8x faster than GenPRM. The *FlexiVe* (Flex@8) configuration is identified as an optimal trade-off point.

**TFLOPS and Wall-Time Efficiency** Figure 4 (left) shows that *FlexiVe* is more efficient. The *FlexiVe* **(Think@k)** configurations define a new state-of-the-art Pareto frontier. For instance, *FlexiVe* (Think@4) attains a higher F1 score (∼87) than the best GenPRM-32B model (∼84) while using less than half the computation (∼12 vs. ∼29 TFLOPS).

Crucially, considering wall-clock time reveals the distinct advantage of the Flex mode over the Think mode. While *FlexiVe* (Think@2) offers competitive TFLOPS efficiency, it requires executing high-latency "Slow Thinking" traces sequentially. In contrast, *FlexiVe* (Flex@8) executes eight low-latency "Fast Thinking" runs in parallel, escalating to slow thinking only when necessary. As shown in Figure 4 (right), this results in drastically different latencies: Flex mode achieves a median wall time of ∼2s (matching the 'NoThink' baseline), whereas Think mode requires ∼18s. Thus, while TFLOPS are comparable, Flex provides a superior accuracy-latency trade-off essential for real-world deployment.

**Optimal Trade-off Analysis and Hyperparameter Selection** The *FlexiVe* **(Flex@8)** configuration, highlighted in the figure, offers an optimal trade-off between cost and performance. It achieves a

substantial F1 score of 78 with a modest computational cost of 7 relative TFLOPS. This analysis provides a basis for hyperparameter selection, as this point represents the "knee" of the performance curve—securing most of the accuracy gains without the high expense of premium 'Think' modes. Given this ideal balance for resource-constrained applications, we adopt the Flex@8 setting for *FlexiVe* in the subsequent experiments involving the full Solve-Detect-Verify pipeline.

## 4.4 SOLVE-DETECT-VERIFY: AN EFFICIENT ALTERNATIVE TO BON RANKING

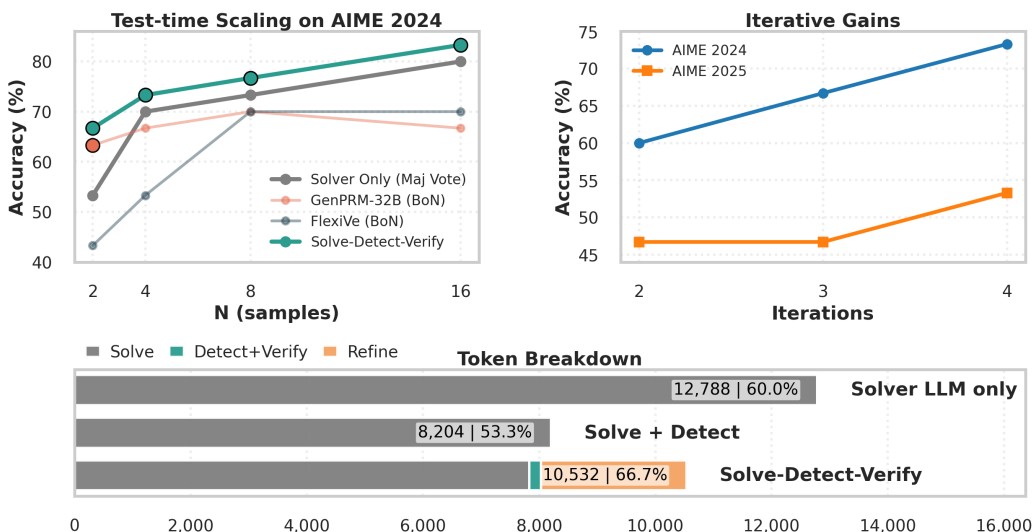

Figure 5: Performance and efficiency analysis of the Solve-Detect-Verify (SDV) pipeline on AIME 2024. **(Top-left)** SDV consistently outperforms standard Best-of-N (BoN) ranking methods in test-time accuracy scaling. **(Top-right)** The iterative nature of SDV yields monotonic accuracy improvements with each refinement step. **(Bottom)** A token breakdown for a single execution of the pipeline (Solve → Detect → Verify) reveals the pipeline's efficiency: the 'Detect' stage reduces token usage, while the 'Verify' stage adds targeted computation to significantly boost accuracy, resulting in a net efficiency gain over the baseline solver.

We evaluate our *Solve-Detect-Verify* framework, demonstrating that its iterative refinement process is a more effective and efficient inference-time strategy than standard Best-of-N (BoN) ranking. The analysis is grounded in performance on the AIME 2024 benchmark.

**Limitations of Standard BoN Ranking** A common scaling strategy, BoN ranking, relies on an external verifier to select the best among $N$ candidate solutions. However, our findings indicate this approach has significant limitations. As shown in Figure 5 (top-left), prominent verifiers like GenPRM-32B struggle to outperform even a simple majority vote baseline. We attribute this to ranking miscalibration, a known issue when verifiers evaluate the lengthy and complex reasoning traces typical of "thinking" models (Wu et al., 2025b). Unlike BoN, which depends on precise scalar scoring for ranking, our Solve-Detect-Verify (SDV) pipeline consistently achieves the highest accuracy across all sample sizes ($N$). At $N = 16$, SDV reaches 83.3% accuracy, surpassing the strong majority vote baseline (80.0%) and substantially outperforming GenPRM-32B BoN (66.7%). This suggests that active iterative self-correction is a more robust scaling mechanism than passive one-shot external ranking.

**The Advantage of Iterative Refinement** The superior performance of SDV is attributable to its iterative refinement mechanism. Unlike BoN, which passively ranks static solutions, SDV actively improves upon them. Figure 5 (top-right) quantifies this benefit, showing a clear, monotonic increase in accuracy with each successive iteration on both the AIME 2024 and 2025 datasets. (Note: Unlike the parallel sampling ($N$) in the left panel, this analysis tracks sequential refinement steps ($T$) on a single solution trajectory.) For AIME 2024, accuracy improves from 60.0% after two iterations to over 70.0% after four, confirming that the refinement process is consistently productive.

**Component-wise Token Efficiency** The SDV pipeline is architected for efficiency, achieving superior accuracy without a corresponding increase in computational cost. The token breakdown in Figure 5

(bottom) provides a detailed analysis. The baseline 'Solver LLM only' approach uses an average of 12,788 tokens. **Detect** stage first prunes unnecessary generation paths, significantly reducing the average token count by over 35% to 8,204. **Verify** stage then applies targeted, corrective feedback, increasing the token count to 10,532 but yielding a substantial accuracy gain from 53.3% to 66.7%. Notably, we observe that the solver generates significantly fewer tokens during refinement compared to the initial phase. We hypothesize that while the base RL training encourages extensive exploration initially, the targeted feedback in the second pass constrains the search space, resulting in more concise corrections. The full SDV pipeline delivers a higher accuracy while consuming approximately 18% fewer tokens than the solver-only baseline, demonstrating a clear net gain in overall efficiency.

### 4.5 Discussions

**Generalizability of Hesitation Detection** We acknowledge that our hesitation keywords were derived empirically. To assess their generalizability, we evaluated the 'Detect' module on models with distinct training paradigms (Table 19). The results indicate that the mechanism's effectiveness is tied to the training method. On RL-distilled models (e.g., Qwen3-8B), the detection behaves predictably, significantly reducing token usage (e.g., -3,576 tokens on AIME 2025) by pruning unproductive paths. Conversely, on SFT-trained models (e.g., S1-14B), the behavior is erratic, often increasing token usage (+2,374 tokens). This suggests that RL training instills a robust link between "verbalized hesitation" and model uncertainty, making our detection strategy a principled approach for the increasingly common class of RL-reasoning models.

Table 3: Sensitivity of Hesitation Keyword Detection Across Training Paradigms. RL-distilled models show consistent token reduction, whereas SFT models exhibit erratic behavior.

| Model (Training Paradigm) | Dataset | Baseline Acc. (%) | Solve+Detect Acc. (%) | Acc. $\Delta(pp)$ | Token $\Delta$ |
|---|---|---|---|---|---|
| Qwen3-8B (RL-distilled) | AIME 2024 | 83.3 | 60.9 | -22.4 | -1,144 |
| | AIME 2025 | 73.3 | 66.7 | -6.6 | -3,576 |
| S1 14B (SFT-trained) | AIME 2024 | 30.0 | 26.7 | -3.3 | +2,206 |
| | AIME 2025 | 13.3 | 33.3 | +20.0 | +2,374 |

**Component Robustness and Qualitative Analysis** Our extended analyses in the appendix validate the key design choices and robustness of our pipeline. The `Flex@k` verifier's dynamic escalation is governed by a consensus threshold ($\tau = 0.8$) that optimally balances accuracy gains with a nearly 8x reduction in token usage compared to its full "slow thinking" mode (Appendix A.3.1, Table 18). Finally, the iterative refinement loop demonstrates practical utility by successfully correcting 25% of incorrect initial solutions on AIME 2024 (Appendix A.3.3). However, qualitative analysis shows that while feedback effectively restructures algebraic problems, its ability to guide corrections in complex geometric reasoning remains a limitation, pointing to clear avenues for future work (Appendix A.3.4).

## 5 Conclusion and Future Work

**Conclusion** We introduce *FlexiVe* , a dynamic verifier balancing computational cost and accuracy, integrated into the *Solve-Detect-Verify* pipeline for efficient LLM reasoning enhancement. Experiments confirm that our pipeline, leveraging *FlexiVe* , achieves significant gains in both accuracy and token efficiency over baselines, highlighting flexible verification and intelligent pipeline design as a scalable path toward more reliable and efficient complex reasoning in LLMs.

**Limitation and Future Work** *FlexiVe* and *Solve-Detect-Verify* opens several exciting avenues for future research. Our empirical validation focuses on the challenging domain of mathematical reasoning, a standard practice for rigorously evaluating complex reasoning frameworks (Zhong et al., 2025; Zhao et al., 2025b; Zheng et al., 2024a; Wang et al., 2024a). A natural and promising next step is to extend the demonstrated benefits of *FlexiVe* to broader domains. This presents a straightforward opportunity to adapt the current "hesitation keywords", an effective heuristic for mathematical traces, to new linguistic patterns. From a systems perspective, the pipeline's computational profile reflects a deliberate trade-off for enhanced verification accuracy. We see a clear path to optimizing this by integrating state-of-the-art inference engines like vLLM (Kwon et al., 2023) or SGLang (Zheng et al., 2024b). These future steps represent a clear roadmap toward evolving our framework into a more general-purpose, highly efficient, and robust system for verified reasoning.

## ETHICS STATEMENT

We have adhered to the ICLR Code of Ethics in the development and evaluation of this research. Our work focuses on improving the reasoning capabilities and inference efficiency of large language models on publicly available mathematical benchmark datasets (`gsm8k`, `math`, `olympiadbench`, and `omnimath`). We acknowledge the dual-use nature of advanced AI problem-solvers; while they can serve as valuable tools for education and research, they could also be misused for academic dishonesty. The goal of our research is to contribute to the scientific understanding of AI reasoning and create more reliable and efficient systems, not to facilitate misuse. Our method, `FlexiVe`, uses pre-trained models without further fine-tuning, and we have made no effort to remove existing safety guards. We believe our work contributes to the transparent and responsible development of AI systems.

## REPRODUCIBILITY STATEMENT

We are committed to the reproducibility of our research. All experiments were conducted using publicly available large language models and standard academic benchmarks, the specifics of which are detailed in the experimental setup section. To facilitate full reproduction of our results, we will make our source code publicly available upon publication. This release will include the implementation of the `FlexiVe` framework, scripts for running the evaluations, and the exact prompts used for generation and feedback (as shown in Figures 6, 7, etc.). Key hyperparameters and experimental settings, such as the number of voting samples ($N$) for each configuration, are described in our results tables (Tables 15-17) and throughout the appendix. Our code is available at `https://anonymous.4open.science/r/flexive-7D5D`.

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

# A APPENDIX

## THE USE OF LARGE LANGUAGE MODELS

We use Large Language Models (LLMs), including ChatGPT and Gemini, solely for the purpose of editing and polishing the writing in this paper.

## BROADER IMPACT

The development of `FlexiVe` and the `Solve-Detect-Verify` pipeline represents a significant step toward making advanced AI reasoning systems more practical, reliable, and efficient. By designing a verifier that dynamically allocates computational resources—switching between rapid "fast thinking" and meticulous "slow thinking"—our framework directly confronts the critical trade-off between accuracy and efficiency that currently limits the deployment of large models. This approach promotes a more sustainable and scalable paradigm for AI reasoning, reducing the reliance on computationally expensive, brute-force methods like Best-of-N sampling with process-based verifiers. Our work has the potential to enhance trust and safety in AI systems. By not only identifying but also pinpointing the exact location of errors and providing targeted feedback for correction, our pipeline improves the interpretability and debuggability of the reasoning process. This iterative refinement is crucial for high-stakes domains where reliability is paramount, such as automated scientific discovery, medical diagnostics, and educational tools. By making state-of-the-art reasoning more computationally accessible, our work also helps democratize advanced AI, enabling powerful capabilities to run in more resource-constrained environments. This research paves the way for future investigations into more sophisticated self-correcting systems and adaptive computation, pushing the frontier of efficient and trustworthy artificial intelligence.

## A.1 IMPLEMENTATION DETAILS AND EXPERIMENTAL SETUP

This section provides comprehensive details regarding the training of `FlexiVe`, the implementation of the `Solve-Detect-Verify` pipeline, evaluation benchmarks, and specific implementation clarifications.

### A.1.1 FLEXIVE TRAINING

**Training Protocol and Rationale** We train `FlexiVe` using Group Relative Policy Optimization (GRPO) (Shao et al., 2024a) initialized from the DeepSeek-R1-Distill-Qwen-14B model (Shao et al., 2024a). We utilize the BIG-Bench Mistake dataset (Tyen et al., 2024), using 1,526 samples for training and 170 for testing, derived from a 90%/10% split. The objective is to predict the first error index ($idx_{gt}$) or -1 if correct, optimized using the composite reward (Section 3.2, main paper). Training initially focused on optimizing the "Fast Thinking" mode (NoThinking activated) to instill efficient, accurate error detection with minimal verbosity. This strategy established a strong, low-cost baseline and promoted data efficiency, providing a robust foundation that generalized well to the "Slow Thinking" mode. Statistics for the training data are provided in Table 4.

Table 4: Details of the model and dataset used for training.

| Items | Values |
| --- | --- |
| Model | FlexiVe-14B |
| Benchmark | BIG-Bench Mistake |
| Train Set Size | 1,526 |
| Test Set Size | 170 |

**RL vs. SFT Generalization** As discussed in the main paper (Section 4.2), our RL approach demonstrated superior generalization compared to Supervised Fine-Tuning (SFT). An SFT baseline trained on 10,000 complex reasoning paths showed poor generalization when evaluated on the diverse, often simpler traces in ProcessBench. In contrast, `FlexiVe`, RL-trained on only 1,526 samples, generalized effectively. This highlights RL's advantage in fostering robust verifiers capable of handling diverse reasoning styles and complexities, even with significantly less data.

**Hyperparameters and Optimization** We employed LoRA (Hu et al., 2022) targeting attention projection layers and used the AdamW (Loshchilov and Hutter, 2019) optimizer with gradient

checkpointing. Training utilized the `transformers` (Wolf et al., 2020) and `trl` (von Werra et al., 2020-2024) libraries, tracked via Weights & Biases (Biewald, 2020). The key hyperparameters and optimization settings are summarized in Table 5.

Table 5: Training details.

| Parameter | Value | Description |
|---|---|---|
| Base Model | DeepSeek-R1-Distill-Qwen-14B | Base model for initialization |
| Learning Rate | $5 \times 10^{-6}$ | Initial learning rate |
| Batch Size | 1 | Per-device batch size |
| Num Train Epochs | 3 | Number of training epochs |
| Gradient Accum. Steps | 8 | Gradient accumulation steps |
| PEFT / LoRA | True (r=16, $\alpha$=32) | Adapter fine-tuning (LoRA) |
| LR Scheduler Type | Linear | Learning Rate Scheduler Type |
| Optimizer | AdamW | Optimization algorithm |
| Warmup Steps | 100 | Number of warmup steps |
| GRPO Group Size | 14 | Number of generations per prompt |
| KL Coefficient | 0.04 | KL penalty coefficient for GRPO |

### A.1.2 EVALUATION BENCHMARKS

We assessed our framework on a suite of challenging mathematical reasoning benchmarks. For evaluating step-level verification performance, we used the four standard splits of the **ProcessBench** benchmark: GSM8K, MATH, Olympiad-Bench, and OmniMATH. For evaluating the end-to-end performance of the full `Solve-Detect-Verify` pipeline, we used problems from the **AIME 2024** and **AIME 2025** competitions. The number of problems in the test set for each benchmark is detailed in Table 6.

Table 6: Details of datasets used for model evaluation.

| Benchmark | Test Set Size |
|---|---|
| *ProcessBench Splits* | |
| GSM8K | 400 |
| MATH | 1,000 |
| Olympiad-Bench | 1,000 |
| OmniMATH | 1,000 |
| *End-to-End Evaluation* | |
| AIME 2024 | 30 |
| AIME 2025 | 30 |

### A.1.3 SOLVE–DETECT–VERIFY PIPELINE

The `Solve-Detect-Verify` pipeline employs an adaptive, iterative strategy. Algorithm 2 outlines the implemented flow, focusing on the iterative refinement process generalized for $T$ attempts.

---

**Algorithm 2** `Solve-Detect-Verify` Pipeline Implementation Flow

---

**Require:** Problem $P$, Verification Parameters $\Theta_V = (k_{fast}, \tau_{agree}, k_{slow})$, Max Attempts $T$
1: $S_{current} \leftarrow$ NIL
2: $F_{prev} \leftarrow$ NIL
3: **for** $t = 1$ to $T$ **do**
4:                                    ▷ — Solve and Detect (Algorithm 1, main paper) —
5:     $Prompt_t \leftarrow$ FormatPrompt$(P, S_{current}, F_{prev})$
6:     $S_t \leftarrow$ GenerateSolutionWithDetection$(LLM, Prompt_t)$
7:     $S_{current} \leftarrow S_t$
8:     **if** $t < T$ **then**                            ▷ Verify if not the last attempt
9:                                              ▷ — Verify (`FlexiVe`) —
10:         (is_valid$_t$, error_step$_t$, $F_t$) $\leftarrow$ AdaptiveVerify$(P, S_t, \Theta_V)$
11:         **if** is_valid$_t$ = True **then**
12:             **break**                            ▷ Solution verified, terminate early
13:         **else**
14:             $F_{prev} \leftarrow F_t$                    ▷ Prepare feedback for refinement
15: **return** $S_{current}$

---

**Solve Module and Prompts** We employ DeepSeek-R1-14B/32B as the solver LLM. The initial prompt (Figure 6) guides the model to generate a structured solution. If refinement is required ($t > 1$), a retry prompt (Figure 7) incorporates feedback from `FlexiVe` ($F_{prev}$).

---

**LLM Initial Solver Prompt**

```
The following is a math problem:
[Math Problem]
{question}
Solve it step by step. For each step, you
should use \n\n in the end.
Please put your final answer (i.e., the
index) in \\boxed{{}}.
```

Figure 6: LLM Initial Solver Prompt.

---

**LLM Retry Prompt with Feedback**

```
The following is a math problem:
[Math Problem] {question}

You previously attempted to solve this:
[Previous Solution]
{previous_solution}

The feedback is:
[Verification Feedback]
{verifier_feedback}

Please correct your solution.
Provide a complete, new solution.
Put your final answer in \\boxed{{}}.
```

Figure 7: LLM Retry Prompt with Feedback.

---

**Detect Module** The `GenerateSolutionWithDetection` function implements a streaming detection framework to identify and curtail "overthinking."

- **Hesitation Keywords:** Generation is monitored for hesitation cues (Figure 8). These keywords were derived empirically by observing common phrases signaling a pause or self-correction in LLM outputs.

- **Completeness Check:** Upon detecting a keyword, the proposer is suspended. A Detector LLM (the same base model) evaluates the context using the prompt in Figure 9. We compare the log-probabilities of "Yes" and "No" to determine completeness.

- **Efficiency (KV Cache Reuse):** The 'Detect' module achieves high efficiency by leveraging vLLM (Kwon et al., 2023) with prefix caching. Since the detection prompt is a continuation of the existing generation context, vLLM automatically reuses the KV cache from preceding steps, leading to minimal overhead (more than 90% reuse).

- **"Continue-after-detected" Logic:** If completeness is detected, the generation might be briefly continued to ensure the current thought segment is fully articulated before truncation, facilitating better context for potential sequential revision.

---

**Hesitation Keywords**

```
Wait, double-check, Alternatively, Hmm,
Let me check, Alright, make sure,
Another way, Let me verify, to confirm,
Looking back, But wait
```

Figure 8: Hesitation keywords monitored for detection.

---

**LLM Detection Prompt**

```
You are a solution completeness checker.
Given current solution to a math problem,
determine if it is a complete solution
(i.e., contains a final answer).
Respond with exactly one word: 'Yes' if
complete, 'No' otherwise.
```

Figure 9: LLM Detection Prompt.

**Verify Module (`AdaptiveVerify`)** This function implements the Flexible Allocation of Verification Budget (Section 3.2). It conducts $k_{fast}$ "Fast Thinking" runs. If the agreement ratio meets $\tau_{agree}$, the consensus is returned. Otherwise, it escalates to $k_{slow}$ "Slow Thinking" runs. Across all experiments, $k_{slow}$ is consistently set to $\lceil \mathbf{k_{fast}/8} \rceil$, balancing cost reduction with sufficient analysis to resolve ambiguities.

### A.1.4 EVALUATION BENCHMARKS AND BASELINES

**`FlexiVe` Evaluation** We assess step-level verification capabilities (F1 score) using ProcessBench (Zheng et al., 2024a) (GSM8K, MATH, OlympiadBench, OmniMATH). We compare against SOTA Process Reward Models (PRMs), including GenPRM (Zhao et al., 2025b) and Dyve (Zhong et al., 2025).

**Pipeline Evaluation** We evaluate the end-to-end effectiveness of the `Solve-Detect-Verify` pipeline on challenging mathematical datasets: AIME (2024, 2025) (Aim, 2024; 2025), AMC, CNMO (Liu et al., 2024) (China's National Mathematical Olympiad), and OlympiadBench. We measure accuracy and efficiency (tokens, TFLOPS). We use DeepSeek-R1 14B/32B (Shao et al., 2024a) as the base worker LLMs, comparing against direct generation and Self-Consistency (Wang et al., 2023).

**Compute Categories** In Table 10, models are categorized by computational effort:

- **Moderate Compute:** Involves a reasonable number of samples without code execution (e.g., GenPRM Maj@8 w/o code, `FlexiVe` Flex@k). The adaptive nature of Flex@k keeps the average compute moderate.

- **High Compute:** Prioritizes maximal accuracy using extensive verification or intensive techniques (e.g., GenPRM Maj@8 w/ Code Exec, `FlexiVe` Think@64).

## A.2 DETAILED EXPERIMENTAL RESULTS

### A.2.1 FLEXIVE PERFORMANCE SCALING

Tables 7 (Think@k), 8 (NoThinking@k), and 9 (Flex@k) provide detailed F1 scores and total token consumption (in Millions, M) for `FlexiVe` across ProcessBench subsets.

Table 7: Performance of `FlexiVe` "With Thinking" (Think@k) on ProcessBench subsets. Tokens are total generated (Millions) across the respective test set.

| | GSM8K | | MATH | | OlympiadBench | | OmniMATH | |
|---|---|---|---|---|---|---|---|---|
| $k$ | F1 (%) | Tokens (M) | F1 (%) | Tokens (M) | F1 (%) | Tokens (M) | F1 (%) | Tokens (M) |
| 2 | 82.3 | 2.4 | 81.9 | 5.2 | 78.0 | 8.4 | 71.3 | 7.1 |
| 4 | 86.7 | 4.8 | 86.4 | 10.4 | 84.3 | 16.8 | 76.9 | 14.3 |
| 8 | 86.4 | 9.5 | 88.9 | 20.9 | 85.4 | 33.4 | 78.9 | 28.6 |
| 16 | 87.6 | 19.2 | 89.7 | 41.8 | 86.5 | 66.9 | 80.1 | 57.1 |
| 32 | 87.7 | 38.1 | 89.7 | 83.8 | 86.7 | 133.6 | 80.6 | 114.2 |
| 64 | 87.8 | 76.3 | 90.1 | 167.5 | 86.7 | 267.3 | 80.4 | 228.4 |
| 128 | 88.1 | 152.7 | 90.0 | 335.4 | 86.7 | 534.1 | 80.5 | 456.4 |

Table 8: Performance of `FlexiVe` "Without Thinking" (NoThinking@k) on ProcessBench subsets.

| | GSM8K | | MATH | | OlympiadBench | | OmniMATH | |
|---|---|---|---|---|---|---|---|---|
| $k$ | F1 (%) | Tokens (M) | F1 (%) | Tokens (M) | F1 (%) | Tokens (M) | F1 (%) | Tokens (M) |
| 2 | 61.5 | 0.4 | 57.2 | 1.5 | 49.0 | 1.9 | 50.5 | 1.6 |
| 4 | 66.8 | 0.7 | 61.3 | 3.0 | 53.8 | 3.7 | 52.5 | 3.3 |
| 8 | 66.7 | 1.5 | 62.8 | 6.1 | 55.2 | 7.5 | 53.6 | 6.6 |
| 16 | 66.8 | 3.0 | 64.3 | 12.1 | 55.9 | 15.0 | 54.2 | 13.3 |
| 32 | 66.5 | 5.9 | 64.4 | 24.2 | 55.9 | 29.9 | 54.7 | 26.5 |
| 64 | 66.8 | 11.8 | 64.2 | 48.5 | 56.1 | 59.8 | 54.0 | 52.9 |
| 128 | 66.7 | 23.7 | 65.0 | 96.8 | 56.3 | 119.8 | 54.1 | 105.9 |

Table 9: Performance of `FlexiVe` with Flexible Allocation (Flex@k) on ProcessBench subsets.

| | GSM8K | | MATH | | OlympiadBench | | OmniMATH | |
|---|---|---|---|---|---|---|---|---|
| $k$ | F1 (%) | Tokens (M) | F1 (%) | Tokens (M) | F1 (%) | Tokens (M) | F1 (%) | Tokens (M) |
| 2 | 72.97 | 0.2 | 72.92 | 1.0 | 67.43 | 1.3 | 61.41 | 1.3 |
| 4 | 78.43 | 0.3 | 77.67 | 1.5 | 72.41 | 2.1 | 67.34 | 2.1 |
| 8 | 75.75 | 0.5 | 78.86 | 3.1 | 70.06 | 4.3 | 66.57 | 4.2 |
| 16 | 76.88 | 0.9 | 78.20 | 6.1 | 73.07 | 8.2 | 68.94 | 7.9 |
| 32 | 82.84 | 2.1 | 83.30 | 13.9 | 79.23 | 19.5 | 73.40 | 18.8 |
| 64 | 82.00 | 4.3 | 83.63 | 28.7 | 79.26 | 39.6 | 74.67 | 38.6 |
| 128 | 83.02 | 8.9 | 84.96 | 59.1 | 79.98 | 80.8 | 75.23 | 78.5 |

**Analysis of Trade-offs and Efficiency** The data demonstrates the distinct trade-offs. Think@k establishes the accuracy upper bound at the highest cost. NoThinking@k is the most efficient but has the lowest accuracy ceiling. Flex@k effectively balances these extremes. On MATH@128, Flex@k (84.96% F1, 59.1M tokens) achieves an 82.4% token reduction compared to Think@128 (90.0% F1, 335.4M tokens).

At $k = 128$, Flex@k uses approximately 86.1% fewer tokens on average than Think@k. Notably, at higher $k$ values, Flex@k can be both more accurate and more token-efficient than NoThinking@k (e.g., on GSM8K and MATH).

**Visualizing F1 Scaling** Figure 10 visualizes the F1 score scaling corresponding to the data above. Flex@k consistently outperforms NoThinking@k and generally matches or exceeds the DS14B baseline, confirming the effectiveness of the adaptive approach.

Figure 10: F1 score scaling with voting budget $k$ on GSM8K (left) and MATH (right). `FlexiVe` (Flex@k, green circles) improves with larger $k$, performing comparably or better than DS14B (blue triangles, baseline verifier), while both surpass the `FlexiVe` (NoThinking variant, red squares).

### A.2.2 COMPREHENSIVE PROCESSBENCH RESULTS

Table 10 provides a comprehensive comparison of `FlexiVe` on ProcessBench. `FlexiVe` demonstrates strong performance and remarkable sample efficiency, achieving SOTA results despite being trained on only 1,526 samples, compared to 23K-404K samples for other models. In the Moderate Compute category, Flex@128 achieves the best average F1 (80.8%). In the High Compute category, Think@64 establishes a new SOTA for open-source models (86.3% Avg F1).

### A.2.3 STATISTICAL SIGNIFICANCE AND STABILITY

To validate robustness, we simulated the voting process 10 times from a pool of 512 cached completions to generate 95% confidence intervals for `FlexiVe`'s performance (Tables 15, 16, and 17, showing selected $k$ values for brevity).

The analysis finds: (1) **Think@k** shows high stability (tight intervals $\leq$1%). (2) **NoThink@k** exhibits higher variance (wider intervals 2-5%). (3) **Flex@k** achieves a balanced trade-off (moderate intervals 1-4%), validating the reliability of the adaptive approach.

### A.2.4 PARETO FRONTIER ANALYSIS DATA (FIGURE 4)

Table 11 provides the detailed data points corresponding to the Pareto frontier analysis presented in Figure 4 (main paper), comparing F1 scores and Relative TFLOPS on the MATH split of Processbench (Zheng et al., 2024a).

### A.2.5 SOLVE-DETECT-VERIFY PIPELINE PERFORMANCE DATA (FIGURE 5)

This section provides the underlying data supporting the analysis presented in Section 4.4 and Figure 5 (main paper), focusing on the AIME 2024 benchmark.

**Scaling Performance (BoN vs. SDV)** Table 12 details the accuracy scaling as the number of samples ($N$) increases. The SDV pipeline consistently outperforms both simple majority voting and BoN ranking using external verifiers.

**Iterative Gains** Table 13 demonstrates the monotonic accuracy improvements achieved through the iterative refinement process of the SDV pipeline.

Table 10: ProcessBench results reported with F1 scores. Results for *FlexiVe* are highlighted . **bold** indicates the best in the sub category. All *FlexiVe* variants are trained on only 1526 samples.

| Model | # Samples | GSM8K | MATH | Olympiad Bench | Omni-MATH | Avg. |
|---|---|---|---|---|---|---|
| *Proprietary Models* | | | | | | |
| GPT-4o-0806 | unk | 79.2 | 63.6 | 51.4 | 53.5 | 61.9 |
| o1-mini | unk | 93.2 | 88.9 | 87.2 | 82.4 | 87.9 |
| *Open Source Models (1.5B)* | | | | | | |
| Skywork-PRM-1.5B | unk | 59.0 | 48.0 | 19.3 | 19.2 | 36.4 |
| GenPRM-1.5B (Pass@1) w/ Code Exec | 23K | 52.8 | 66.6 | 55.1 | 54.5 | 57.3 |
| *Open Source Models (7-8B)* | | | | | | |
| Math-Shepherd-PRM-7B | 445K | 47.9 | 29.5 | 24.8 | 23.8 | 31.5 |
| RLHFlow-PRM-Mistral-8B | 273K | 50.4 | 33.4 | 13.8 | 15.8 | 28.4 |
| EurusPRM-Stage2 | 30K | 47.3 | 35.7 | 21.2 | 20.9 | 31.3 |
| Qwen2.5-Math-PRM-7B | ∼344K | 82.4 | 77.6 | 67.5 | 66.3 | 73.5 |
| RetrievalPRM-7B | 404K | 74.6 | 71.1 | 60.2 | 57.3 | 65.8 |
| Universal-PRM-7B | unk | 85.8 | 77.7 | 67.6 | 66.4 | 74.3 |
| Direct Generative PRM-7B | 23K | 63.9 | 65.8 | 54.5 | 55.9 | 60.0 |
| GenPRM-7B w/ Code Exec (Pass@1) | 23K | 78.7 | 80.3 | 72.2 | 69.8 | 75.2 |
| GenPRM-7B w/ Code Exec (Maj@8) | 23K | 81.0 | 85.7 | 78.4 | 76.8 | 80.5 |
| *Open Source Models (14-32B) w/ **Moderate Compute*** | | | | | | |
| Dyve-14B | 117K | 68.5 | 58.3 | 49.0 | 47.2 | 55.8 |
| GenPRM-32B w/o Code Exec (Maj@8) | 23K | 78.8 | 85.1 | 78.7 | 74.9 | 79.3 |
| *FlexiVe* (Flex@32) | 1526 | 82.8 | 83.3 | 79.2 | 73.4 | 79.7 |
| *FlexiVe* (Flex@128) | 1526 | **83.0** | **85.0** | **80.0** | **75.2** | **80.8** |
| *Open Source Models (14-32B) w/ **High Compute*** | | | | | | |
| GenPRM-32B (Pass@1) w/ Code Exec | 23K | 83.1 | 81.7 | 72.8 | 72.8 | 77.6 |
| GenPRM-32B (Maj@8) w/ Code Exec | 23K | 85.1 | 86.3 | 78.9 | 80.1 | 82.6 |
| *FlexiVe* (Think@64) | 1526 | **88.1** | **90.1** | **86.7** | **80.4** | **86.3** |

Table 11: Detailed Data for Pareto Frontier Analysis on MATH Dataset (Figure 4).

| Model | Config (@k) | F1 Score (%) | Relative TFLOPS |
|---|---|---|---|
| FlexiVe (Flex) | @2 | 72.9 | 1.9 |
| | @4 | 75.8 | 3.8 |
| | @8 (Best Trade-off) | 78.9 | 7.5 |
| | @16 | 78.2 | 13.2 |
| | @32 | 83.3 | 27.2 |
| FlexiVe (Think) | @1 | 81.9 | 6.1 |
| | @2 | 82.3 | 10.2 |
| | @4 | 86.4 | 12.3 |
| | @8 | 88.9 | 22.8 |
| GenPRM-7B (Maj) | @1 | 80.3 | 15.1 |
| | @8 | 83.1 | 28.1 |
| GenPRM-32B (Maj) | @1 | 80.0 | 13.4 |
| | @8 | 83.5 | 29.4 |

**Token Efficiency Breakdown** Table 14 details the average token usage and accuracy at each stage of the pipeline, illustrating the efficiency gains from the 'Detect' stage and the accuracy boost from the 'Verify' stage.

### A.2.6 SCALING PROPERTIES

We explore scaling *Solve-Detect-Verify* along two dimensions: the verifier budget (Flex@N) and the solver budget (Number of Solutions).

Table 12: Test-time Accuracy Scaling on AIME 2024 (Data supporting Figure 5, Top-Left).

| Method | N=2 | N=4 | N=8 | N=16 |
|---|---|---|---|---|
| Solver Only (Maj Vote) | 53.3 | 70.0 | 73.3 | 80.0 |
| GenPRM-32B (BoN) | 63.3 | 66.7 | 70.0 | 66.7 |
| FlexiVe (BoN) | 43.3 | 53.3 | 70.0 | 70.0 |
| **Solve-Detect-Verify** | **66.7** | **73.3** | **76.7** | **83.3** |

Table 13: Iterative Refinement Gains (Data supporting Figure 5, Top-Right).

| Iterations | AIME 2024 Accuracy (%) | AIME 2025 Accuracy (%) |
|---|---|---|
| 2 | 60.0 | 46.7 |
| 3 | 66.7 | 46.7 |
| 4 | 73.3 | 53.3 |

Table 14: Token Efficiency Breakdown on AIME 2024 (Data supporting Figure 5, Bottom).

| Configuration | Average Tokens | Accuracy (%) |
|---|---|---|
| Solver LLM only | 12,788 | 60.0 |
| Solve + Detect | 8,204 | 53.3 |
| Solve-Detect-Verify | 10,532 | 66.7 |

*Scaling Verifier Budget (Flex@N):* We analyze scaling *FlexiVe* 's budget within a single pipeline run (Figure 11). The 'w/o Flex' setup significantly cuts token usage (e.g., 0.67 ratio on AIME2024) but reduces accuracy. Integrating 'Flex@8' substantially boosts accuracy over the baseline (e.g., 73.3% vs. 56.6% on AIME2024) while still using fewer tokens (0.96 ratio).

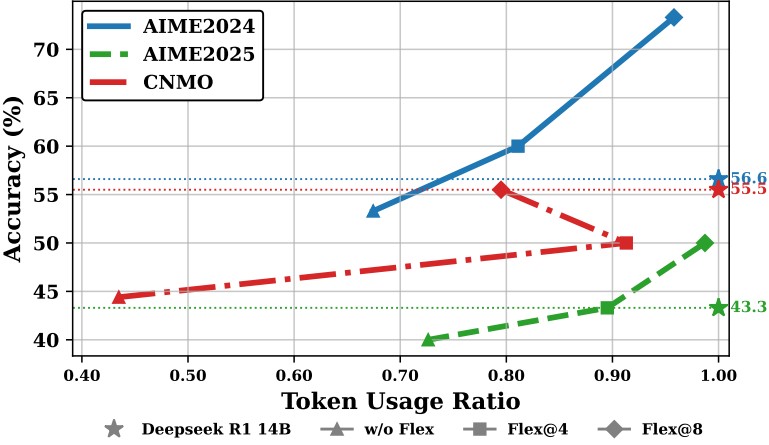

Figure 11: Impact of scaling *FlexiVe* 's verification budget (Flex@N) within a single *Solve-Detect-Verify* execution on Pass@1 Accuracy vs. Token Usage Ratio relative to DeepSeek R1 14B.

*Scaling Solver Budget:* To achieve higher peak accuracies, we scale compute by generating multiple solutions from the solver. On AIME2024 (Figure 5, top left panel), this strategy yields significant improvements, climbing from 67.5% (1 solution) to over 83% (16 solutions), requiring approximately 4x fewer solutions than the baseline to reach similar accuracy levels.

### A.2.7 FLEXIVE PERFORMANCE SCALING DETAILS

This section provides a more detailed breakdown of the performance scaling for the different configurations of our `FlexiVe` method. We present the 95% confidence intervals for accuracy on four benchmark datasets as the number of voting samples ($N$) increases.

The results are detailed for the "With Thinking" configuration (`Think@k`) in Table 15, the "Without Thinking" configuration (`NoThink@k`) in Table 16, and our primary `FlexiVe` method (`Flex@k`) in Table 17.

A consistent trend is evident across all tables: performance generally improves as the number of voting samples ($N$) increases from 2 to 128. For example, for the main `Flex@k` method on the `math` dataset, accuracy climbs from 72.9% to 85.0%. Concurrently, the confidence intervals tend to narrow with a larger $N$, indicating more stable and reliable results. These tables also quantitatively show that the `Think@k` method consistently achieves the highest performance, while `NoThink@k` establishes a performance baseline.

Table 15: 95% Confidence Intervals for `FlexiVe` "With Thinking" (Think@k).

| Voting N | gsm8k | math | olympiadbench | omnimath |
|---|---|---|---|---|
| 2 | $82.3 \pm 0.89$ | $81.9 \pm 0.67$ | $78.0 \pm 0.38$ | $71.3 \pm 0.56$ |
| 8 | $86.4 \pm 0.50$ | $88.9 \pm 0.21$ | $85.4 \pm 0.40$ | $78.9 \pm 0.19$ |
| 32 | $87.7 \pm 0.44$ | $89.7 \pm 0.24$ | $86.7 \pm 0.33$ | $80.6 \pm 0.21$ |
| 128 | $88.1 \pm 0.32$ | $90.0 \pm 0.15$ | $86.7 \pm 0.15$ | $80.5 \pm 0.09$ |

Table 16: 95% Confidence Intervals for `FlexiVe` "Without Thinking" (NoThink@k).

| Voting N | gsm8k | math | olympiadbench | omnimath |
|---|---|---|---|---|
| 2 | $61.5 \pm 2.36$ | $57.2 \pm 4.28$ | $49.0 \pm 2.56$ | $50.5 \pm 3.55$ |
| 8 | $66.7 \pm 2.63$ | $62.8 \pm 4.91$ | $55.2 \pm 3.32$ | $53.6 \pm 4.05$ |
| 32 | $66.5 \pm 2.35$ | $64.4 \pm 4.96$ | $55.9 \pm 2.98$ | $54.7 \pm 3.93$ |
| 128 | $66.7 \pm 2.46$ | $65.0 \pm 5.09$ | $56.3 \pm 3.23$ | $54.1 \pm 4.08$ |

Table 17: 95% Confidence Intervals for `FlexiVe` (Flex@k).

| Voting N | gsm8k | math | olympiadbench | omnimath |
|---|---|---|---|---|
| 2 | $73.0 \pm 2.74$ | $72.9 \pm 4.08$ | $67.4 \pm 2.62$ | $61.4 \pm 3.34$ |
| 8 | $75.8 \pm 2.48$ | $78.9 \pm 2.85$ | $70.1 \pm 2.05$ | $66.6 \pm 2.57$ |
| 32 | $82.8 \pm 1.17$ | $83.3 \pm 2.38$ | $79.2 \pm 1.41$ | $73.4 \pm 2.39$ |
| 128 | $83.0 \pm 1.32$ | $85.0 \pm 1.48$ | $80.0 \pm 1.51$ | $75.2 \pm 2.47$ |

## A.3 EXTENDED DISCUSSIONS AND ANALYSIS

### A.3.1 SENSITIVITY ANALYSIS OF CONSENSUS THRESHOLD $\tau$

The consensus threshold $\tau$ governs the escalation from "Fast Thinking" to "Slow Thinking" in the Flex@k strategy. We performed a sensitivity analysis (Table 18) to validate our choice of $\tau = 0.8$.

Table 18: Sensitivity analysis for the consensus threshold $\tau$ in Flex@8, averaged across ProcessBench datasets. As $\tau$ varies, performance shifts between the 'NoThink@8' and 'Think@8' baselines.

| Consensus Threshold ($\tau$) | Slow Thinking Escalation (%) | Avg. F1 Score (%) | Avg. Total Tokens (M) |
|---|---|---|---|
| *NoThink@8 Baseline* | *0%* | *59.6* | *5.4* |
| 0.5 | 5% | 61.0 | 5.2 |
| 0.7 | 18% | 69.5 | 4.1 |
| **0.8 (Chosen)** | **28%** | **72.9** | **3.0** |
| 0.95 | 80% | 83.5 | 19.5 |
| *Think@8 Baseline* | *100%* | *84.9* | *23.1* |

At a low threshold ($\tau = 0.5$), escalation is minimal (5%), and performance approaches the 'No-Think@8' baseline. At a strict threshold ($\tau = 0.95$), the system escalates 80% of cases, approaching the 'Think@8' baseline but at a massive computational cost. Our chosen value, $\tau = 0.8$, represents the optimal balance, significantly raising the F1 score (72.9%) while maintaining high efficiency (3.0M tokens, nearly 8x lower than Think@8).

### A.3.2 ROBUSTNESS OF THE DETECTION MECHANISM

As discussed in Section 4.5 (main paper), the robustness of the hesitation keyword detector depends on the model's training paradigm (Table 19). On RL-distilled models (e.g., Qwen3-8B), the mechanism behaves predictably. However, on SFT-trained models (e.g., S1 14B), its behavior is erratic, sometimes increasing token usage and causing unpredictable accuracy shifts. This suggests RL instills a more reliable link between hesitation keywords and model uncertainty.

Table 19: Sensitivity of Hesitation Keyword Detection Across Training Paradigms. (Table 2 in main paper).

| Model (Training Paradigm) | Dataset | Baseline Acc. (%) | Solve+Detect Acc. (%) | Acc. $\Delta(pp)$ | Token $\Delta$ |
|---|---|---|---|---|---|
| Qwen3-8B (RL-distilled) | AIME 2024 | 83.3 | 60.9 | -22.4 | -1,144 |
| | AIME 2025 | 73.3 | 66.7 | -6.6 | -3,576 |
| S1 14B (SFT-trained) | AIME 2024 | 30.0 | 26.7 | -3.3 | +2,206 |
| | AIME 2025 | 13.3 | 33.3 | +20.0 | +2,374 |

### A.3.3 EFFECTIVENESS OF THE REFINEMENT LOOP

We analyzed the refinement success rate on the AIME 2024 dataset. Out of 16 initial solutions that were incorrect (S1), our pipeline successfully corrected 4 of them (S2), yielding a **25% success rate**. This demonstrates the practical utility of the refinement mechanism, particularly noteworthy as `FlexiVe` was not fine-tuned on the solver's specific traces, indicating good generalization.

### A.3.4 QUALITATIVE ANALYSIS OF THE FEEDBACK MECHANISM

We analyzed successful and failed feedback attempts to provide deeper insight into the correction process.

---

**Successful S1 → S2 Correction**

**Problem:** Every morning Aya goes for a 9-kilometer-long walk... When she walks at a constant speed of $s$... the walk takes her 4 hours, including $t$ min...
**S1 Error at Step:** 2
**FlexiVe Feedback (Excerpt):** ...Understanding the problem: Aya walks 9 km at two different speeds... We need to find the total time when she walks at $(s + \frac{1}{2})$ km/h.
Setting up equations:

- First scenario: $4 = \frac{9}{s} + \frac{t}{60}$

- Second scenario: $2.4 = \frac{9}{s+2} + \frac{t}{60}$

Subtracting equations: ...
**Result: S2 was correct**

---

> ### Ineffective Feedback (Failed Correction)
>
> **Problem:** Let B be the set of rectangular boxes with surface area 54 and volume 23. Let $r$ be the radius of the smallest sphere that can contain...
>
> **Error Location by FlexiVe:** Step 13
>
> **FlexiVe Feedback (Excerpt):** The solution starts by understanding that the radius... is half the space diagonal... $r^2 = \frac{a^2+b^2+c^2}{4}$. The goal is to maximize $a^2 + b^2 + c^2$ given the constraints...
>
> **Outcome: Correction failed**
>
> **Analysis:** For complex geometry problems, `FlexiVe` may fail to produce a corrective pathway, highlighting a limitation in advanced spatial and geometric problem-solving capabilities.

