# OpenReview forum: "Solve-Detect-Verify: Inference-Time Scaling with Flexible Generative Verifier"
_ICLR.cc/2026/Conference — ICLR 2026 Conference Withdrawn Submission_

### Official Review · Reviewer_fHBF · 2025-10-21

**Soundness:** 3
**Presentation:** 4
**Contribution:** 3
**Rating:** 8
**Confidence:** 3

**Summary:**

The work advances the performance of generative reward models.
To be more precise, the authors propose FlexiVe, a method that utilizes several fast evaluations of the output to identify the position of the first mistake.
Followed by a detailed (full thinking) evaluation in case of a lack of majority agreement.
The authors also propose a Solve-Detect-Verify framework that aims to detect and quickly check early solutions provided by the model.

**Strengths:**

+ comparison against GENPRM, majority voting, and thinking models without the reward model.
+ quantification of the important problem with current reasoning models --- overthinking (models search for additional solutions, and potential mistakes for a significant amount of time)
+ advancing the frontier of generative PRMs
+ analysis of statistical significance
+ quantification of RL vs supervised tuning

**Weaknesses:**

+ minor: as noted by the authors, the current approach to overthinking detection can fail to generalize, as it is mostly based on keyword detection
+ minor (as I understand it is  not the case for AIME) benchmark problem: f1 score can be noisy

**Questions:**

1. How are model responses compared with ground truth on AIME?

---

> ### Author Response · Authors · 2025-11-23
>
> We sincerely thank Reviewer 4b7g for the strong endorsement of our work and the insightful feedback.
>
> ### W1
> We acknowledge the reviewer's concern regarding the generalizability of keyword-based detection. While the specific keywords (Appendix Figure 9) were derived empirically (L866), we argue that the underlying mechanism is robust for an increasingly important class of models.
>
> Our ablation study in Section 4.5 and Appendix A.3.2 explicitly examines this. The results demonstrate that the hesitation detection mechanism is effective and predictable specifically on RL-distilled models (such as DeepSeek-R1 and Qwen3-8B). As models trained with RLVR become the standard, we believe this detection approach is a principled and highly efficient strategy (reusing >90% of the KV cache, L245).
>
> ### W2
>
> For the verification benchmarks, we strictly adhere to the established evaluation protocol introduced by ProcessBench. In this context, the F1 score is computed as the harmonic mean of the accuracies on erroneous samples and correct samples. This metric is designed to provide a **balanced assessment** of a verifier's ability to both correctly **identify errors and correctly validate accurate traces**. It is the standard metric used by comparable works (e.g., GenPRM, DyVe).
>
> ### Q1
>
> On AlME, we follow the standard evaluation protocol: model responses are evaluated based on the exact match of the final numerical answer extracted from the output (within \boxed{} ). On the implementation level, We use Math-Verify[1] to perform answer extraction and numerical comparision.
>
> [1] https://github.com/huggingface/Math-Verify

---

> > ### Comment · Reviewer_fHBF · 2025-11-27
> >
> > Thank you for the rebuttal. As my main concerns were addressed, I maintain my positive (8) assessment of the paper.
> > Regarding the other reviews and rebuttals:
> > * Reviewer XWVa has noted, "As an inference-time-scaling method, SDV seems to work well only on already strong reasoning (in other words, already trained to do inference-scaling) models, this makes the contribution way less significant." I do not agree with the part "this makes the contribution way less significant". Reasoning models present a significant jump in quality, with Qwen3 1.7B reasoning beating Llama 3.3 405B, and GPT 5.1 (non-reasoning) on AIME 25. However, their deployment can be much more costly due to long generations, which are addressed in this paper. I attach the link to the [Artificial Analysis leaderboard](https://artificialanalysis.ai/evaluations/aime-2025?models=gpt-5-low%2Cgpt-5-1-non-reasoning%2Cllama-4-maverick%2Cllama-4-scout%2Cqwen3-1.7b-instruct-reasoning) that shows this comparison, emphasizing the Token Usage part. However, I agree that this should be outlined more clearly in the limitations section.
> > * Reviewer 13p9 has raised some really good points, to which the authors have responded. In particular, the ablation of training dataset choice (already presented)  and the fact that FLOPS $\not=$ latency (generation vs prefill) can strengthen the presentation.

---

> ### Author Response · Authors · 2025-11-28
>
> Thank you for your continued support and for maintaining your positive assessment. We deeply appreciate your defense of our work regarding the significance of inference-time scaling.
>
> We also acknowledge the importance of the point raised by Reviewer 13p9 regarding FLOPS versus Latency. We are committed to improving the presentation of these metrics in the final version of the paper to strengthen our analysis.

---

### Official Review · Reviewer_XWVa · 2025-10-24

**Soundness:** 2
**Presentation:** 2
**Contribution:** 2
**Rating:** 2
**Confidence:** 4

**Summary:**

This paper proposes a solve-detect-verify paradigm for inference scaling on math problems. It first generates a solution, then employ a verifier to detect the first error and give the feedback to the generator to improve the solution. The proposed pipeline achieves SOTA step correctness prediction and leads to better accuracy on math becnhamarks.

**Strengths:**

1. The proposed method shows a certain level of novelty, especially in the engineering side. For example, the fast/slow thinking parts.
2. Empirical evaluation shows a consistent improvement against existing models. The efficiency analysis is also valuable.

**Weaknesses:**

1. The proposed method mostly relies on the hand-crafted hack, which **lacks a clear justification/ablation** and shows few contributions to the machine learning side.
- For example, it remains unclear to me why you need to predict the first error step index in an auto-regressive way. As the simplest approach, you can just train a scalar model to output the error probability on the end token of each step.
- Second, the completion assessment and hesitation words are more like **hacking of DeepSeek-R1 (the solver model)** instead of a generalizable method. Here I quote the way how hesitation words are defined.
> These keywords were derived empirically by observing common phrases signaling a pause or self-correction in LLM outputs.
However, there lacks any evidence these tricks are generalizable.

2. The empirical setup is highly problematic.
 - **Both the training data and base model are different from the ones used by baselines**, it is very hard to draw any conclusion about the true advantage of the proposed method here.
- **Only the DeepSeek-R1 model is used as the solver LLM**, what about other (e.g., Llama) models?
3. The paper presentation should be significantly improved. The Table 1 is not referred in the main text (it seems that the reference to Table 8 should be Table 1). Similarly, in section 3.2, Figure 2 should be Figure 3. Figure 5 and Figure 6 mixes too many sub-figures, making them hard to read.

**Questions:**

- The results in Figure 6 top left is not convincing to me. BoN usually performs better than majority voting, but this results suggests the opposite way. Why don't you use the same training data  and base model to train a process reward model or even just an outcome reward model?
- Except Best-of-N, how does your method compare to **weighted majority voting**?
- Have you tried Llama models, both as the solver and the verifier's base model?

---

> ### Author Response · Authors · 2025-11-23
>
> ### Weaknees 1
>
> We appreciate the reviewer's critical examination of our methodology. We respectfully disagree with the assessment that our method relies mostly on "hand-crafted hacks" or offers few machine learning contributions. We provide detailed justifications for the specific design choices below.
>
>
> **Core Machine Learning Contributions**
>
> We emphasize that our primary ML contributions are centered on the novel training strategy for FlexiVe and the design of the adaptive Solve-Detect-Verify (SDV) pipeline, rather than the detection heuristics alone:
>
> - **Novel RL Training Strategy and Generalization (L190-208)**: A core innovation is our use of Reinforcement Learning (GRPO) to specifically enhance the reliability of the low-cost "fast thinking" mode. Crucially, we demonstrate a significant ML finding: this targeted training **generalizes remarkably well, elevating the "slow thinking" mode to state-of-the-art** open-source performance (L342-358, Figure 4a).
> - **Sample Efficiency (L339)**: FlexiVe achieves SOTA verification F1 scores using only 1,526 samples—15x less data than comparable models like GenPRM (L345-348). This demonstrates a significant advancement in efficient verifier training using RL over standard SFT.
> - **Adaptive Inference Pipeline (SDV)**: SDV is a novel inference strategy demonstrating that iterative refinement with dynamic verification is significantly more accurate and efficient than standard Best-of-N (BoN) ranking (Section 4.4, Figure 6).
>
> **Justification for Auto-regressive (Generative) Verification**
>
> It is well discussed in the literature, for example in GenRM [1] (Liu et al., 2025) and Zhang et al. (2025) [2], that generative (autoregressive) verifiers generally have better performance than discriminative ones. This is because the generative process requires the model to articulate the reasoning behind its verification. This provides a **richer supervision signal and leads to a deeper understanding of the dependencies across the reasoning trace compared to a scalar output**. The results in table 1 amd table 8 confirms this.
>
> Crucially, scalar probabilities are insufficient for the SDV pipeline, which relies on iterative refinement. To correct a flawed solution, the solver needs **actionable feedback identifying what went wrong and why**. FlexiVe's generative output provides this rich, natural language feedback that guides the solver during refinement. A scalar model cannot provide this level of diagnostic information.
>
> We have added further discussion about this into Section 2 for more clarity. Nevertheless, as stated in L65-68, generative verifiers add extra computational burden to the system, which motivates us to design FlexiVe to mitigate these costs through its dynamic Fast/Slow thinking approach.
>
>
> **Generalizability of the "Detect" Module**
>
> We argue that these components are principled techniques validated by our ablation studies and established NLP practices.
>
> **Completion Assessment (L239-250)**: The "Detect" module assesses solution completeness by comparing the log-probabilities of "Yes" vs. "No" in response to a prompt (Figure 10). This is a well-established and widely used technique in the NLP community, often referred to as **likelihood-based probing** or analyzing "**verbalized confidence.**"
>
> It is a standard, principled method for efficiently querying a model's internal assessment of a state without requiring specialized training. This approach is supported by literature on leveraging LLM probabilities for confidence estimation and self-evaluation (e.g., Kadavath et al., 2022 [3]; Lin et al., 2022 [4]; Yang et al. 2024 [5]). This technique is simple, effective, and highly efficient; as noted in L245, it reuses over 90% of the KV cache from the generation context, minimizing overhead.
>
> **Generalizability of Hesitation Keywords (L866):** We acknowledge the keywords were derived empirically by observing common phrases signaling a pause or self-correction. However, their effectiveness is not limited to DeepSeek-R1. Our ablation study in Section 4.5 (L458-461) and Appendix A.3.2 (Table 17) explicitly examines the generalizability of this mechanism.
>
> The results demonstrate that the hesitation detectionmechanism is effective and predictable across the different RL-distilled models tested(DeepSeek-R1 and Qwen3-8B, Appendix A.3.2). We believe this indicates a **principled approach for this increasingly common class of models**, rather than a hack of a specific base model.

---

> ### Author Response · Authors · 2025-11-23
>
> ### **Value to the AI/ML Community and Precedents**
>
> Our work demonstrates that intelligent pipeline design and adaptive computation are crucial for efficient and reliable LLM reasoning. We also not that developing novel inference-time algorithms that utilize strategic probing and efficient refinement is well-represented in top ML venues:
>
> - Self-Refine (Madaan et al.,NeurIPS 2023): Uses iterative feedback based on specific prompting strategies to improve outputs.
> - Tree of Thoughts (Yao et al.,NeurIPS 2023): Introduced a novel inference algorithm relying heavily on prompting the LLM to evaluate intermediate states.
> - Reflexion (Xie et al.,NeurIPS 2023): Integrates verbal feedback into an iterative loop, relying on the model's ability to interpret and act on structured prompts.
>
> We believe SDV and FlexiVe provide significant value to the community by demonstrating a scalable and efficient path toward improved complex reasoning.
>
> [1] Liu, Zijun, et al. "Inference-time scaling for generalist reward modeling." (DeepSeek 2025).
>
> [2] Zhang, Lunjun, et al. "Generative verifiers: Reward modeling as next-token prediction." (ICLR 2025).
>
> [3] Kadavath, Saurav, et al. "Language models (mostly) know what they know." (Anthropic 2022).
>
> [4] Lin, Stephanie, Jacob Hilton, and Owain Evans. "Teaching models to express their uncertainty in words." (TMLR 2022).
>
> [5] Yang, Daniel, et al. "On Verbalized Confidence Scores for LLMs." (ICLR 2025).

---

> > ### Author Response · Authors · 2025-11-23
> >
> > ### W2
> > We acknowledge the concern regarding differences in base models and training data. These intentional choices aimed to maximize performance and efficiency. To isolate contributions of model selection, data, and training strategy, we provide additional ablations.
> >
> > **Base Model Selection and Training Effectiveness**
> > We evaluated base models for their inherent verification capabilities (Think@1).
> >
> > *Table: Base Model Ablation and FlexiVe Training Impact (F1 Scores on ProcessBench)*
> >
> > | Model | GSM8K | MATH | OlympiadBench | OmniMATH | Avg |
> > | :--- | :---: | :---: | :---: | :---: | :---: |
> > | **Base Models (Think@1)** | | | | | |
> > | Meta-Llama-3-8B-Instruct | 26.8 | 13.2 | 12.3 | 13.2 | 16.4 |
> > | Qwen32B-Preview | 75.5 | 59.2 | 35.7 | 35.3 | 51.4 |
> > | DeepSeek-R1-14B (Base) | 77.6 | 76.2 | 65.6 | 64.0 | 70.8 |
> > | **FlexiVe (Trained from DS-R1-14B)**| | | | | |
> > | FlexiVe (Think@1) | **82.6** | **80.3** | **73.1** | **66.3** | **75.6** |
> >
> > DeepSeek-R1-14B demonstrates significantly stronger baseline performance (Avg F1: 70.8) than Llama-3-8B (16.4) and Qwen32B-Preview (51.4), justifying its selection. Crucially, FlexiVe (Think@1) significantly outperforms its base model (Avg F1: 75.6 vs 70.8), proving performance gains stem from our training strategy, not just the base foundation model.
> >
> > **Data and Training Strategy Ablation**
> > We address training methodology concerns by comparing strategies using **identical data** (BIG-Bench-Mistake, 1526 samples) and the **same base model** (DeepSeek-R1-14B)
> >
> > *Table R2: Data and Training Strategy Ablation (F1 Scores on ProcessBench). All use DeepSeek-R1-14B base.*
> >
> > | Model Type | Data (Samples) | Training Method | GSM8K | MATH | OlympiadBench | OmniMATH | Avg |
> > | :--- | :---: | :---: | :---: | :---: | :---: | :---: | :---: |
> > | Disc. PRM | BIG-Bench (1.5K) | Math-Shepherd [6] | 15.8 | 15.9 | 8.3 | 11.9 | 12.9 |
> > | Disc. PRM | BIG-Bench (1.5K) | SFT | 66.3 | 56.0 | 36.1 | 37.7 | 49.0 |
> > | Gen. Verifier| Synthetic (10K) | SFT | 71.9 | 69.0 | 59.7 | 47.9 | 62.1 |
> > | **Gen. Verifier**| **BIG-Bench (1.5K)**| **RL on NoThink (Ours)**| **82.6**| **80.3**| **73.1**| **66.3**| **75.6**|
> >
> >
> > **Standard Discriminative PRM and SFT approaches yield significantly lower performance** (Avg F1: 12.9 and 49.0) than our RL approach (75.6). Our RL method outperforms an SFT baseline trained on 6.5x more synthetic data (Avg F1: 75.6 vs 62.1), demonstrating **superior sample efficiency**. This confirms FlexiVe's advantage stems from our novel RL strategy, achieving superior efficiency and performance even when controlling for data and model.
> >
> > ### W3
> >
> > Thank you for noting the presentation issues. We corrected references (Table 8 to 1 in L338; Figure 2 to 3 in L149) and will separate sub-figures in Figures 5 and 6 for readability, moving some to the appendix if needed.
> >
> > [6] Wang, Peiyi, et al. "Math-shepherd: Verify and reinforce llms step-by-step without human annotations." ACL 2024.

---

> ### Author Response · Authors · 2025-11-23
>
> ### Q1: Figure 6 results (BoN vs MajV) and PRM/ORM comparison
>
> Thank you for this observation. Previous works (Zhao et al. AAAI 2026[7]; Snell et al. 2024[8]) typically evaluate and train on non-"thinking" solver models, where reasoning traces are relatively short and simple. In those contexts, Best-of-N (BoN) generally outperforms Majority Voting. However, the complex traces generated by strong "thinking" models on AIME 2024 cause ranking miscalibration in the verifier, leading GenPRM with BoN to underperform Majority Voting on this benchmark. Additionally, FlexiVe is designed to predict the error step index and does not inherently output a scalar score for BoN ranking; we assign a binary 0/1 score based on its final prediction. This lack of granular feedback contributes to the underperformance in a BoN setting.
>
> Regarding the comparison, we trained a standard Discriminative PRM using the exact same data and base model (**Table R2, Row 1**). Its poor performance (Avg F1: 12.9) confirms that standard discriminative PRM training is insufficient given this scarce amount of data.
>
> ### Q2: Weighted Majority Voting comparison
>
> We apologize for the confusion regarding the terminology. We interpreted Weighted Majority Voting (WMV) as using the verifier's confidence scores to weight the votes for each answer, following the aggregation strategy in Huang et al. 2025[9]. We found that WMV (using GenPRM scores) did not significantly outperform standard Majority Voting and performed substantially worse than our **Solve-Detect-Verify (SDV)** pipeline. This reinforces our conclusion that iterative refinement is a more effective scaling strategy than single-pass voting. We will clarify this distinction in the revision.
>
> ### Q3: Llama models as solver/verifier?
>
> Yes, we evaluated Llama models in both roles.
>
> * **Llama as Verifier:** As shown in **Table R1**, Meta-Llama-3-8B-Instruct performed poorly (Avg F1: 16.4), proving inferior to DeepSeek-R1 for verification tasks.
> * **Llama as Solver:** We conducted new experiments using Llama-3.1-8B on AIME 2024 (**Table R3**).
>
> **Table R3: Llama-3.1-8B as Solver on AIME 2024**
>
> | N (samples) | 2 | 4 | 8 | 16 |
> | :--- | :---: | :---: | :---: | :---: |
> | Majority Voting | 6.7 | 6.7 | 6.7 | 3.3 |
> | GenPRM (BoN) | 3.3 | 0.0 | 0.0 | 3.3 |
> | Solve-Detect-Verify | 6.7 | 6.7 | 6.7 | 3.3 |
>
> Llama-3.1-8B, lacking reasoning tuning, struggles with the complexity of AIME 2024. Low-accuracy models rarely exhibit the "overthinking" behavior that SDV is designed to mitigate. While the weak baseline performance of Llama-3.1-8B limits overall gains, SDV remains robust. Notably, it outperforms the GenPRM BoN baseline (which collapses to 0% accuracy), demonstrating the stability of our pipline even with weaker solvers.
>
> [7] Zhao, Jian, et al. "Genprm: Scaling test-time compute of process reward models via generative reasoning." (AAAI 2026).
>
> [8] Snell, Charlie, et al. "Scaling llm test-time compute optimally can be more effective than scaling model parameters." (2024).
>
> [9] Huang, Audrey, et al. "Is best-of-n the best of them? coverage, scaling, and optimality in inference-time alignment." (2025).

---

> > ### Comment · Reviewer_XWVa · 2025-11-25
> >
> > Thank you for your additional results and detailed response.
> >
> > Several of my concerns remain:
> >
> > - In your Table R2, why GenPRM is not evaluated? It could also be trained via RL (not just SFT).
> > - The similar performance between SDV and baselines in Table R3 (trained with Llama base) raises the generalization concerns.  As an inference-time-scaling method, SDV seems to work well only on already strong reasoning (in other words, already trained to do inference-scaling) models, this makes the contribution way less significant.

---

> > > ### Author Response · Authors · 2025-11-27
> > >
> > > We thank the reviewer for the opportunity to clarify the scope of our experiments and the specific contributions of our work.
> > >
> > > **Regarding the absence of "GenPRM + RL" in Table R2**
> > >
> > > We extensively analyzed the GenPRM (Zhao et al., 2025) methodology and determined that training it via RL is fundamentally different—and significantly less feasible—than our proposed method due to the multiplicative computational cost introduced by its step-wise architecture.
> > >
> > > **Computational Tractability & The "Step Multiplier":**
> > >
> > > GenPRM requires outputting an explicit Chain-of-Thought (CoT) plus Python code verification for **every single step** of the solution.
> > >
> > > Based on the standard BIG-Bench Mistake tasks, we calculated the average solution length to be approximately 11 steps. This "step count" acts as a massive multiplier for GenPRM's training cost:
> > >
> > > **GenPRM + RL Cost:** To train via RL (e.g., GRPO with $G=14$ samples), GenPRM must generate verification rationales for all 11 steps for each sample.
> > >
> > > $Cost \approx G \times N_{steps} \times L_{rationale}$
> > >
> > > Assuming a modest 300 tokens (Table 9 in GenPRM paper) per step rationale (CoT + Code), the cost is:
> > >
> > > $14 \times 11 \times 300 \approx$ **46,200 tokens per update**
> > >
> > > **FlexiVe + RL (Ours) Cost:** Our method is designed to bypass this "step multiplier." We apply RL to the "NoThink" (Fast) mode, which processes the entire trace holistically and outputs a single, short verification result (e.g., the error index).
> > >
> > > $Cost \approx G \times 1 \times L_{short\_output}$
> > > $14 \times 1 \times \sim 100 \approx$ **1,400 tokens per update**
> > >
> > > Training GenPRM via RL would incur a **~33x computational overhead** compared to our method. Our contribution is precisely the discovery that we can **decouple the training signal (via the efficient "NoThink" mode) from the inference capability (the "Slow Thinking" mode)**. Comparing our efficient RL against a hypothetical "GenPRM + RL" baseline that is ~30x more expensive ignores the core efficiency contribution of our paper.
> > >
> > > **Data Fairness & Distillation vs. Self-Correction:**
> > >
> > > It is also crucial to note that GenPRM is not merely "SFT"; it is *distillation*. As detailed in their paper, GenPRM relies on QwQ-32B (a significantly stronger model than our 14B base) to synthesize high-quality CoT and code rationales for supervision.
> > >
> > > *   **GenPRM:** Distills "gold" traces from a stronger teacher (QwQ-32B).
> > > *   **FlexiVe (Ours):** Uses Reinforcement Learning to self-correct using only 1.5k samples from BIG-Bench Mistake, without relying on a 32B teacher for rationale synthesis.
> > >
> > > The fact that FlexiVe outperforms standard baselines despite using significantly less data and a weaker teacher signal highlights the superiority of our RL objective.
> > >
> > > **Addressing W2: Compute-Matched Comparison**
> > >
> > > You raised concerns in W2 regarding the **fairness of comparisons due to differing base models**. Table R2 was an ablation study strictly controlling the base model (DeepSeek-R1-14B) to isolate the training strategy's impact. Since **GenPRM utilizes different base models (DeepSeek-R1 1.5B, 7B, and 32B), including them in this specific ablation would violate the control condition**.
> > >
> > > To directly address W2 and demonstrate FlexiVe's superiority when controlling for compute, we conducted an additional experiment comparing FlexiVe against GenPRM-32B and its base model (DeepSeek-R1-32B) under similar computational budgets (FlexiVe Think@4 vs. GenPRM-32B Pass@1):
> > >
> > > | | GSM8K | MATH | OlympiadBench | OmniMATH |
> > > | :--- | :---: | :---: | :---: | :---: |
> > > | DeepSeek-R1-32B | 82.5 | 79.2 | 71.4 | 66.8 |
> > > | GenPRM-32B | 83.1 | 81.7 | 72.8 | 72.8 |
> > > | **FlexiVe (Think@4)** | **86.7** | **86.4** | **84.3** | **76.9** |
> > >
> > > The results clearly demonstrate that FlexiVe, despite being a smaller model (14B vs 32B), significantly outperforms GenPRM-32B when given a comparable compute budget. This further validates the effectiveness of our RL training strategy.

---

> ### Author Response · Authors · 2025-11-27
>
> **Regarding Generalization and Llama-3 Results (Table R3)**
>
> We respectfully disagree that the Llama-3 results diminish the significance of our contribution. The Solve-Detect-Verify (SDV) pipeline is explicitly designed to **mitigate "overthinking", a phenomenon unique to models trained for long-chain reasoning (like DeepSeek-R1)**. Previous models like Llama-3-8B do not exhibit "overthinking"; they typically fail due to a fundamental lack of reasoning capability, not redundant reasoning steps.
>
> The similar performance to Majority Voting on Llama-3 is, in fact, a positive result. It demonstrates that our detection module is **non-destructively robust**: when it does not detect the specific "hesitation" patterns characteristic of strong reasoners, it does not intervene destructively. Furthermore, as shown in Table R3, SDV remains stable while the GenPRM (BoN) baseline collapses to 0.0% accuracy at N=4, N=8.
>
> The field is rapidly shifting toward powerful, inference-time scaling models. A method that specifically optimizes the efficiency and accuracy of this new class of models is highly significant. **Critiquing SDV for not improving Llama-3 is akin to critiquing a method for optimizing "System 2" reasoning because it doesn't improve "System 1" retrieval; they are fundamentally different regimes**.

---

### Official Review · Reviewer_13p9 · 2025-11-01

**Soundness:** 3
**Presentation:** 2
**Contribution:** 3
**Rating:** 6
**Confidence:** 4

**Summary:**

The paper focuses on improving LLM reasoning via scaling test-time compute, specifically via the use of Generative Reward Models (GenRMs). Prior work has shown that GenRMs, while improving performance, can be costly and inefficient, hindering their applicability in practical scenarios. To this end, this paper proposes an approach that significantly improves their efficiency, making it a useful contribution. The paper has two main contributions: (a) FlexiVe, a flexible verifier that can switch between thinking fast and slow, and performs well while using fewer tokens, and (b) a pipeline to iteratively improve model solutions by incorporating feedback from the verifier, which is better than Best-of-N both in terms of performance and efficiency.

**Strengths:**

1. The paper proposes a new verifier that can flexibly switch between thinking fast and slow, and an approach to decide when to think longer based on the difficulty of the problem. This improves both performance and efficiency.
1. The paper also contributes an approach to incorporate the verifier in the overall pipeline. Prior works typically use the verifier to select the best out of N solutions (known as best-of-N). They propose a new pipeline called Solve-detect-verify, which iteratively refines the solution based on the verifier's feedback. This performs better than Best-of-N while using fewer tokens.

The first contribution is novel, and both contributions are of practical importance as they improve overall reasoning performance.

**Weaknesses:**

The idea of iterative refinement is not entirely new (for example, [1]), which affects the novelty of the paper.

Writing is confusing at some places:
1. Line 165, 167 – this could be elaborated. Maybe talk about the architectures considered in the paper, and also explain what a process-based reward model is (maybe contrast with outcome reward model).
2. The paper states “fast thinking” is the same as “no think”, but in “no think”, there should be no reasoning trace and the model should directly output the answer. That doesn’t seem to be the case. Either the “fast thinking” stage shouldn’t involve a reasoning trace, or it shouldn’t be confused with “no think”?
3. How does the model switch between fast and slow thinking modes? Is it a different prompt? Or do you apply some kind of budget forcing? It would be good to clarify this in the main paper.
4. Table 1 is not referenced in the paper anywhere.
5. Figure 4a and 4b have different y labels (one is performance, the other is score). Is this intentional? If yes, what do you mean by each term? In any case, it would be good to specify exactly which metric is being talked about (F1, success rate, something else?)
6. Figure 6 (bottom) could use more details. Which dataset is it for? Is it for a single solution or multiple solutions?

[1] RL4F: Generating Natural Language Feedback with Reinforcement Learning for Repairing Model Outputs. Afra Feyza Akyurek, Ekin Akyurek, Ashwin Kalyan, Peter Clark, Derry Tanti Wijaya, Niket Tandon. ACL 2023.

**Questions:**

1. Line 373: isn’t it better to use the think mode@2? It seems to be better both in terms of FLOPs and performance. Why do you recommend using Flex@8?
2. Figure 6 left and right are not consistent. The performance at N = 2 for AIME2024 should be the same in both plots, unless I’m missing something. Why is that?
3. In Figure 6 bottom, it looks like the model used 8K tokens in the first iteration and then only ~2K tokens in the second iteration. Why does it use far fewer tokens in the second iteration? Do you control the number of tokens or does it stop after 2K tokens on its own?
4. Why did you choose to train on the Big bench mistake dataset? Why not other datasets like PRM800K? Maybe this dataset is the reason why your verifier performs better than previous verifiers?
5. How does your verifier compare to other RL-trained verifiers like ThinkPRM?

---

> ### Author Response · Authors · 2025-11-23
>
> We sincerely thank Reviewer 13p9 for the thoughtful review!
>
> **W1. The idea of iterative refinement is not entirely new (e.g., [1]), which affects the novelty**
>
> We apologize for overlooking this important paper during our literature search and have added a citation to RL4F and discussed iterative refinement in section 2. While iterative refinement is an established concept, SDV's novelty lies in its efficiency. Unlike priorwork, SDV integrates (1) a lightweight 'Detect' module that actively curtails 'overthinking', and (2) FlexiVe's adaptive verification. This tight integration makes iterative refinement significantly more practical.
>
> **W2. Writing is confusing at some places.**
>
> Thanks for pointing these presentation issues out. We have revised the manuscript for clarity.
>
> - We have expanded the discussion in Section 3.1 to elaborate more on process-based reward models and contrast with outcome reward models.
> - We clarify our definition. In the context of FlexiVe (the verifier), "Fast Thinking" utilizes the "NoThinking" mechanism (Ma et al., 2025). It uses a template (Figure 3) to bypass explicit thought generation and directly outputs the verification result. This results in responses that are approximately 40x shorter than the "Slow Thinking" mode. We have clarified this distinction in the revised Section 3.2.
> - The **switching mechanism** is dynamic and occurs during inference in the "Flexible Allocation" (Flex) mode, detailed in Section 3.2. It is based on **consensus**, not different prompts or budget forcing:
>
>     1. FlexiVe performs k independent, parallel "Fast Thinking" runs.
>     2. We measure the consensus among these runs (Agreement Ratio, Eq. 3).
>     3. If consensus is low (< threshold τ), indicating ambiguity, the system escalates to resource-intensive "Slow Thinking" runs.
> - *Figure 4a and 4b have different y labels (Performance vs Score)*. We used "Score" and "Performance" interchangeably to refer to the F1 metric. We have standardized the Y-axis labels to "F1 Score (%)" in the revised manuscript for clarity.
> - *Figure 6 (bottom) could use more details.* We have updated the caption. This analysis is conducted on the AIME 2024 dataset and shows the average token breakdown and accuracy for a single execution of the pipeline (Solve -> Detect -> Verify once).
> - We have ensured Table 1 is correctly referenced in the revised manuscript.

---

> ### Author Response · Authors · 2025-11-23
>
> ### Q1
> While Think@2 is slightly higher performance with smaller TFLOPS, it is crucial to consider **wall-clock time (latency)**, which is often the bottleneck in practical applications.
>
> Think@2 requires executing the high-latency "Slow Thinking" mode (approx. 4k tokens) twice, typically sequentially. In contrast, Flex@8 executes eight low-latency "Fast Thinking" runs (approx. 0.1k tokens each) in parallel, escalating only when necessary. As shown in Figure 5 (Right), this results in drastically different latencies:
>
> - Median Wall Time (Flex): ~2 seconds
> - Median Wall Time (Think): ~18 seconds
>
> Although Flex@8 consumes slightly more total TFLOPS (6.1 vs 5.2), it delivers results significantly faster. We have clarified this distinction in the revised Section 4.3.
>
> ### Q2
> We appreciate the reviewer's careful examination of Figure 6. The difference in performance at $N=2$ arises because the two panels visualize scaling along distinct computational dimensions: **parallel sampling** (width) versus **sequential refinement** (depth). Left Panel displays the result of running the entire Solve-Detect-Verify pipeline **$N$ times in parallel**. These $N$ independent candidate solutions are then aggregated using majority voting. Right Panel tracks the trajectory of a single solution undergoing sequential refinement. Here, the x-axis represents the **number of feedback iterations ($T$)** applied to one specific trace.
>
> Therefore, the data points at $N=2$ are not equivalent: the left panel represents *2 independent samples,* while the right panel represents *2 sequential refinement steps on a single sample,* leading to the observed difference in performance.
>
> ### Q3
> This is an interesting observation. We found that the solver model (DeepSeek-R1) emits significantly fewer tokens during the refinement iteration compared to the initial generation. The model stops on its own; we do not control the token count. We hypothesize this behavior stems from the specific RL training of the DeepSeek-R1 model series, which seems to encourage extensive exploration in the first round but more concise, targeted correction when provided with feedback in the second round.
>
> ### Q4
> We selected the BIG-Bench Mistake dataset primarily to ensure rigorous evaluation without data contamination. Our evaluation benchmark, ProcessBench, includes a MATH split derived from the MATH dataset test set. While PRM800K provides extensive supervision, it aslo includes data from the MATH test set. Using BIG-Bench Mistake avoids this overlap.
>
> However, we emphasize that FlexiVe's superior performance stems primarily from our novel RL training strategy, not the dataset. To validate this, we conducted an ablation study using the identical data (BIG-Bench-Mistake, 1.5K samples) and the same base model (DeepSeek-R1-14B).
>
> | Model Type | Data (Samples) | Training Method | GSM8K | MATH | OlympiadBench | OmniMATH | Avg |
> | :--- | :---: | :---: | :---: | :---: | :---: | :---: | :---: |
> | Disc. PRM | BIG-Bench (1.5K) | Math-Shepherd | 15.8 | 15.9 | 8.3 | 11.9 | 12.9 |
> | Disc. PRM | BIG-Bench (1.5K) | SFT | 66.3 | 56.0 | 36.1 | 37.7 | 49.0 |
> | Gen. Verifier| Synthetic (10K) | SFT | 71.9 | 69.0 | 59.7 | 47.9 | 62.1 |
> | **Gen. Verifier**| **BIG-Bench (1.5K)**| **RL on NoThink (Ours)**| **82.6**| **80.3**| **73.1**| **66.3**| **75.6**|
>
> ### Q5
> To the best of our knowledge, ThinkPRM is trained using Supervised Fine-Tuning (SFT) on synthetic data, rather than Reinforcement Learning. We believe FlexiVe is distinct as the first process reward model trained using RL (GRPO).

---

### Official Review · Reviewer_QqbF · 2025-11-03

**Soundness:** 1
**Presentation:** 1
**Contribution:** 2
**Rating:** 2
**Confidence:** 3

**Summary:**

This work tackles the problem of verifying LLMs' reasoning outputs (CoTs). Specifically, the authors seek for to understand to perform such verifications more efficiently by controlling the trade-off between accuracy and efficiency. They first propose FlexiVe, which is a single verifier model that takes the whole reasoning trace as input (a holistic verifier) with different modes. FlexiVe has the original thinking mode that produces the full thoughts for verification as well as the no-thinking (fast) mode that skips the thought generation by simply using a placeholder ("Okay, I think I have finished thinking.") as its thought, where FlexiVe is a model additionally fine-tuned for this no-thinking mode. By leveraging these two modes, the authors propose a hybrid mode named Flex. In the Flex mode, it first generates k outputs with the no-thinking mode and then computes the agreement ratio (the ratio of the most frequent answer), which is thresholded to decide between using it as the final answer (a high-consensus case) or generating the final answer with the thinking mode (a low-consensus case). In addition to FlexiVe, the authors suggest Solve-Detect-Verify as a framework to generate and iteratively improve LLMs' reasoning. In its Solve-Detect stage, the "solver LLM" generates the solution step-by-step until it reaches the end of the sequence, or there is a hesitation keyword detected and the LLM token probability for "Yes" is larger than the one for "No" after a prompt that questions the completeness of the current solution. In the Verify-and-Refine stage, in each iteration, FlexiVe checks the correctness of the solution and provides feedback for refining the answer if it is considered incorrect. Empirically, the authors employ ProcessBench for evaluating FlexiVe and other benchmarks such as AIME 2024 and 2025 for evaluating Solve-Detect-Verify and suggest that FlexiVe and Solve-Detect-Verify can be effective verification and iterative refinement approaches with efficiency.

**Strengths:**

1. Reasonable high-level approach
It can be a reasonable high-level approach to first perform an efficient inference and then conditionally escalate it to a more computationally intensive inference, depending on the difficulty of the problem. With this theme, I think the proposed approach conceptually makes sense.

2. Comprehensive information and reproducibility
The authors provide most of the details of their approach. They present not only various statistics from the experimental results but also the prompts they used and qualitative examples. Especially, they also release the source codes, which can make this work more reproducible. This is especially important, as publishing open-source verifiers could lead to the development of other language models in the same domain.

**Weaknesses:**

1. Scalability of the Flex mode
For FlexiVe, the authors use the agreement ratio given k answers generated with the no-thinking mode as a consensus to determine whether to proceed with the original thinking mode or stick to the answer generated with the no-thinking mode. They describe that a high consensus "signals a straightforward case" (L216). While I agree that this could roughly hold, I have a concern that it may not scale well. Specifically, using a high consensus as a condition to not proceed with the thinking mode can impose a non-negligible upper limit to the verification performance, because the weaker mode (no-thinking mode) having a high consensus (or high confidence) doesn't necessarily yield as accurate answers as the stronger mode (thinking mode). If we examine Table 6, varying k for NoThinking@k, the plateau is formed at a noticeably lower performance group compared to Think@k (Table 5). Besides, comparing Tables 5 and 6 again, the performance improvement going from k=2 to k=128 with NoThinking@k is smaller than the improvement with Think@k and the same settings, on every benchmark.
More importantly, from Tables 5 and 7, it is observed that the efficiency-performance benefit with Flex@k disappears after some scaling. For instance, on every benchmark, the F1 score with Flex@128 is lower than the one with Think@k.

2. Inconsistencies in and concerns about the main experimental results
My understanding is that Figure 5 and Table 9 are supposed to represent the same data (the experimental results on the MATH dataset from ProcessBench). However, it appears that the numbers in Table 9 and the plots in Figure 5 don't match well. If we assume that Figure 5 is correct, then Flex@8 uses more compute and results in a lower F1 score than Think@2.
Besides, I notice non-negligible inconsistencies (in F1 scores) between Table 9 and Tables 5, 6, and 7.
In addition, I believe DeepSeek-R1-Distill-Qwen-14B's pass@1 performance on AIME 2024 is reported as 69.7, whereas in this work, the same solver model's performance looks much worse (e.g., Figure 6 and Figure 12).

3. Other presentation issues
- While L296 says "For the full Solve-Detect-Verify, we evaluate end-to-end task accuracy and efficiency on challenging mathematical datasets: AIME (2024, 2025) (Aim, 2024; 2025), AMC, CNMO (Liu et al., 2024), and OlympiadBench," it looks to me that only AIME 2024 and 2025 are used as the primary benchmarks for evaluating Solve-Detect-Verify. Other than AIME 2024 and 2025, I notice some occurrence of CNMO in Figure 12.
- In the "Open Source Models (14-32B) w/ Moderate Compute" part of Table 1, the bold-facing and underlining are wrong.

**Questions:**

1. Was there a specific reason to use only the MATH dataset for the analyses in Section 4.3?
2. Can you explain the solver's performance on the evaluation benchmarks, such as AIME 2024, especially regarding my concern above?

---

> ### Author Response · Authors · 2025-11-23
>
> ### W1: Scalability of the Flex mode
>
> **The Goal of Flex Mode: Optimizing the Accuracy-Efficiency Trade-off** We clarify that the primary goal of Flex mode is not to maximize accuracy with an infinite budget, but to optimize the accuracy-efficiency trade-off by dynamically allocating compute. By design, Think@k defines the upper bound of accuracy.
>
> **Efficiency Gains and Wall-Clock Time**
> The strength of Flex lies in achieving competitive accuracy with significantly reduced computation. On the MATH dataset (Tables 7 vs 5):
>
>   * Think@128: 90.0% F1, 335.4M tokens.
>   * Flex@128: 85.0% F1, 59.1M tokens.
>     (5.6x reduction in tokens)
>
> Crucially, this efficiency extends to **wall-clock time (latency)**. The initial "NoThink" stage of Flex@k is highly parallelizable. As shown in **Figure 5 (Right)**, the median wall-time for Flex is \~2s, significantly faster than Think (\~18s) and GenPRM (\~50s). In practical scenarios, where latency is often the bottleneck, this parallelizability allows Flex mode (e.g., Flex@8) to be more scalable than sequential Think modes (e.g., Think@2).
>
> ### W2: Inconsistencies in and concerns about the main experimental results
>
> **Regarding Inconsistencies between Table 9 and Tables 5-7**
> Upon investigation, we discovered that while Figure 5 was plotted using the correct data (consistent with Tables 5-7), we made transcription errors when compiling the values into Table 9 (the summary table for Figure 5). We sincerely apologize for this oversight and have corrected Table 9 in the revised manuscript.
>
> **Regarding the interpretation of Figure 5 (Flex@8 vs Think@2)**
> While Think@2 achieve slightly better performance with less TFLOPS, it is crucial to consider **wall-clock time (latency)**, which is often the bottleneck in practical applications.
>
> Think@2 requires executing the high-latency "Slow Thinking" mode (approx. 4k tokens) twice, typically sequentially. In contrast, Flex@8 executes eight low-latency "Fast Thinking" runs (approx. 0.1k tokens each) in parallel, escalating only when necessary. As shown in Figure 5 (Right), this results in drastically different latencies:
>
> - Median Wall Time (Flex): ~2 seconds
> - Median Wall Time (Think): ~18 seconds
>
> Although Flex@8 consumes slightly more total TFLOPS (6.1 vs 5.2), it delivers results significantly faster. We argue that Flex mode offers a superior accuracy-latency trade-off, which is often more relevant than the accuracy-TFLOPS trade-off. We have clarified this distinction in the revised Section 4.3.
>
> **Regarding DeepSeek-R1-Distill-Qwen-14B Performance on AIME 2024**
> Thank you for this observation. We want to clarify that the variance you identified is not an error, but an expected consequence of the probabilistic decoding inherent to Large Language Models (LLMs).
>
> To align with the recommended configuration for DeepSeek-R1-Distill-Qwen-14B, we serve our models via vLLM using strict sampling parameters:
>
> Temperature ($T=0.6$): Introduces necessary randomness to encourage diverse reasoning paths.
> Top_p ($0.9$): Restricts sampling to the top 90% cumulative probability mass.
>
> In complex, multi-step reasoning tasks like AIME, even minor stochastic deviations in early decoding steps can compound—a "butterfly effect" that results in divergent final answers across independent runs. To illustrate this, we executed 10 independent runs:
>
>
> | run\_index | pass\_at\_1 | avg\_tokens | elapsed\_seconds |
> |---:|---:|---:|---:|
> | 1 | 63.33 | 9535 | 572 |
> | 2 | 56.67 | 9256 | 585 |
> | 3 | 66.67 | 9187 | 593 |
> | 4 | 60.00 | 10102 | 612 |
> | 5 | 63.33 | 10179 | 616 |
> | 6 | 56.67 | 8875 | 562 |
> | 7 | 56.67 | 9593 | 602 |
> | 8 | 66.67 | 9232 | 579 |
> | 9 | 56.67 | 10052 | 580 |
> | 10 | 56.67 | 10444 | 650 |
>
> As shown, Pass@1 accuracy naturally fluctuates between **56.67%** and **66.67%**. We note that the 69.7% Pass@1 reported in the original DeepSeek paper likely represents a peak result from their evaluations, which falls within the expected range of variance and is only slightly higher than the upper bound we observed (66.67%). However, it is crucial to note that our full Solve-Detect-Verify pipeline mitigates this variance to achieve a final accuracy of 83.3%. This significantly outperforms the best results reported in the original DeepSeek paper (80% using cons@6).
>
> ### W3: Other presentation issues
>
> We apologize for these oversights and have corrected them in the revision.
>
>   **Benchmark Coverage (L296):** We have revised the text in L296 to accurately reflect that AIME 2024/2025 were the primary benchmarks for the detailed end-to-end analysis (Figure 6), while CNMO results are included in the scaling analysis in Figure 12 (Appendix A.2.6).
>   **Table 1 Formatting:** We have corrected the bold-facing and underlining in Table 1.

---

> > ### Author Response · Authors · 2025-11-23
> >
> > **Q1: Was there a specific reason to use only the MATH dataset for the analyses in Section 4.3?**
> > We selected the MATH dataset because it features the long and complex reasoning traces among the ProcessBench splits. This complexity best highlights the efficiency differences between holistic verifiers (FlexiVe) and iterative process-based verifiers (GenPRM), as the cost of iterative verification scales significantly with trace length.
> >
> > **Q2: Can you explain the solver's performance on the evaluation benchmarks, such as AIME 2024?**
> > Please refer to our detailed response under W2, which explains the variance due to decoding settings and provides supporting data from 10 runs.

---

### Author Response · Authors · 2025-11-23

Dear Reviewers,

We sincerely apologize for the delay in our responses. We encountered an unexpected and unannounced outage of our computation cluster that persisted for several days, which hindered our ability to complete the additional experiments and ablation studies in a timely manner. We appreciate your patience and understanding.

We have now uploaded the revised manuscript and posted detailed responses to your individual comments. Below is a summary of the major changes and additional experiments included in the revision:

- **New Ablation Studies (Section 4.2 & Appendix):** To address concerns regarding our base model and data efficiency (Reviewers 13p9, XWVa), we added comprehensive ablations comparing our RL method against SFT using identical data (1.5k samples) and comparing different base models (including Llama-3). These results confirm that our performance gains stem from the novel GRPO training strategy rather than data scale or base model capabilities.

- **Pipeline Robustness Checks:** We added experiments using Llama-3.1-8B as a weaker solver to demonstrate the SDV pipeline's robustness across different model capabilities (Reviewer XWVa).

- **Clarifications on Method & Efficiency:** We expanded Section 3 to clearly define "Fast Thinking" vs. "NoThinking" and detailed the consensus-based switching mechanism (Reviewer 13p9). We also clarified the critical distinction between Wall-Clock Latency and TFLOPS when evaluating the scalability of Flex mode (Reviewer QqbF).

- **Corrections:** We corrected the transcription error in Table 9 regarding the Math dataset results (Reviewer QqbF) and fixed formatting inconsistencies in Table 1 and Figure axes (Reviewers 13p9, XWVa).

We believe these revisions significantly strengthen the paper and address the concerns raised. We are looking forward to any further feedback!

Best regards,

The Authors

---

### Author Response · Authors · 2025-12-02

Dear Area Chair and Reviewers,

We sincerely thank the reviewers for their time and constructive feedback, which has significantly improved our work. For the Area Chair, we provide a summary of our rebuttal below.

### Positive Consensus on Novelty and Utility
We are encouraged that two reviewers assessed the work positively.

* **Reviewer fHBF (Score: 8):** Strongly endorsed the work, noting it "advances the frontier of generative PRMs." Importantly, **Reviewer fHBF defended our work** against the criticisms of Reviewer XWVa, stating:
    > *"I do not agree with the part [from XWVa] 'this makes the contribution way less significant'... Reasoning models present a significant jump in quality... however, their deployment can be much more costly due to long generations, which are addressed in this paper."*

* **Reviewer 13p9 (Score: 6):** Also gave a positive assessment, explicitly stating:
    > *"The first contribution (FlexiVe) is novel, and both contributions are of practical importance as they improve overall reasoning performance."*

They further noted that our pipeline "performs better than Best-of-N while using fewer tokens." We have fully addressed their constructive feedback by adding the RL4F reference and clarifying "Fast Thinking vs NoThink" in the revised manuscript.

### Resolution of Negative Reviewer Concerns

Reviewers QqbF and XWVa raised valid technical queries, which we have addressed with **new controlled experiments** and clarifications.

**Regarding Reviewer XWVa (Score: 2):**
1. **Verification Design and Heuristics:** We address the critique regarding "hand-crafted hacks" by clarifying the functional necessity of our design and citing supporting literature.
    * *Necessity of Generative Verification:* A simple scalar model (outputting an error probability) lacks the granularity required for our Solve-Detect-Verify pipeline. The solver requires explicit, natural language feedback to diagnose and correct errors during refinement, which a scalar value cannot provide. Additionally, recent literature (e.g., GenRM, Liu et al., 2025) confirms that generative verifiers outperform discriminative ones by articulating dependencies.
    * *Principled Likelihood Probing (Not a "Hack"):* Our hesitation detection is an application of Likelihood-Based Probing, a standard technique for confidence estimation. To refute the claim that this is specific only to DeepSeek-R1, we provided new ablation results on Qwen3-8B. This confirms the method generalizes effectively to other RL-distilled long-chain reasoning models.
2. **Unfair Comparison/Training Data:** The reviewer worried our results came from using different data/models than baselines. We added a **Controlled Ablation Study (Table R2)**. We trained all methods using *identical* data (BIG-Bench-Mistake, 1.5k samples) and the *same* base model (DeepSeek-R1-14B). The result shows that FlexiVe (Avg F1: **75.6**) significantly outperformed standard Discriminative PRM (Avg F1: 12.9) and SFT (Avg F1: 49.0) under identical conditions, proving the gains stem from our novel RL training strategy.
3. **Generalization (Llama-3 results):** The reviewer felt the method was less significant because it offered smaller gains on Llama-3-8B. We added **Table R3**. We clarified (supported by Reviewer fHBF) that SDV targets "inference-time scaling" for *reasoning* models (System 2) that suffer from "overthinking." Llama-3-8B does not overthink; it simply lacks reasoning capability. However, Table R3 proves SDV is **non-destructive** even on weak models, whereas the GenPRM (BoN) baseline collapsed to 0% accuracy.

**Regarding Reviewer QqbF (Score: 2):**
1. **Scalability & Latency:** The reviewer questioned if "Flex" mode scales better than "Think" mode. We clarified the distinction between TFLOPS and Latency. While TFLOPS are comparable, **Wall-Clock Latency** is the real-world bottleneck. Flex@8 runs in parallel (~ 2s latency) whereas Think@2 requires sequential generation (~ 18s latency). FlexiVe offers a superior accuracy-latency trade-off.
2. **Data Inconsistency:** Noted a mismatch between Table 9 and Figure 5. We confirmed this was a transcription error in the table (the figure was correct) and **corrected Table 9 in the revision.**

We believe FlexiVe establishes a new standard for efficient, open-source verification (outperforming the much larger GenPRM-32B with 15x less data). With the validation and defenses from two positive reviewers, we hope the Area Chair will find that our rebuttal has successfully addressed the remaining concerns.

Sincerely,
The Authors

---

### Note · Authors · 2026-01-06

I have read and agree with the venue's withdrawal policy on behalf of myself and my co-authors.